# PTEN and DNA-PK determine sensitivity and recovery in response to WEE1 inhibition in human breast cancer

Andrä Brunner[1†], Aldwin Suryo Rahmanto[1†], Henrik Johansson[2‡], Marcela Franco[3‡], Johanna Viiliäinen[1], Mohiuddin Gazi[1], Oliver Frings[2], Erik Fredlund[2], Charles Spruck[4], Janne Lehtiö[2], Juha K Rantala[5], Lars-Gunnar Larsson[3], Olle Sangfelt[1]*

[1]Department of Cell and Molecular Biology, Karolinska Institutet, Stockholm, Sweden; [2]Cancer Proteomics Mass Spectrometry, Science for Life Laboratory, Department of Oncology-Pathology, Karolinska Institutet, Stockholm, Sweden; [3]Department of Microbiology, Tumor and Cell biology, Karolinska Institutet, Stockholm, Sweden; [4]Tumor Initiation and Maintenance Program, NCI-Designated Cancer Center, Sanford Burnham Prebys Medical Discovery Institute, La Jolla, United States; [5]Department of Oncology and Metabolism, University of Sheffield, Sheffield, United Kingdom

*For correspondence:
Olle.Sangfelt@ki.se

[†]These authors contributed equally to this work
[‡]These authors also contributed equally to this work

Competing interests: The authors declare that no competing interests exist.

**Abstract** Inhibition of WEE1 kinase by AZD1775 has shown promising results in clinical cancer trials, but markers predicting AZD1775 response are lacking. Here we analysed AZD1775 response in a panel of human breast cancer (BC) cell lines by global proteome/transcriptome profiling and identified two groups of basal-like BC (BLBCs): 'PTEN low' BLBCs were highly sensitive to AZD1775 and failed to recover following removal of AZD1775, while 'PTEN high' BLBCs recovered. AZD1775 induced phosphorylation of DNA-PK, protecting cells from replication-associated DNA damage and promoting cellular recovery. Deletion of DNA-PK or PTEN, or inhibition of DNA-PK sensitized recovering BLBCs to AZD1775 by abrogating replication arrest, allowing replication despite DNA damage. This was linked to reduced CHK1 activation, increased cyclin E levels and apoptosis. In conclusion, we identified PTEN and DNA-PK as essential regulators of replication checkpoint arrest in response to AZD1775 and defined PTEN as a promising biomarker for efficient WEE1 cancer therapy.

## Introduction

Deregulation of DNA replication, often attributed to aberrant oncogene-induced replication stress (RS) is a hallmark and driver of cancerogenesis (*Hanahan and Weinberg, 2011*). RS induces various DNA lesions, including highly cytotoxic DNA double-strand breaks (DSBs), leading to an increased reliance on replication checkpoints and repair activities and has therefore been described as an ¨Achilles heel¨ of cancer (*Bartkova et al., 2006*; *Kotsantis et al., 2018*; *Maya-Mendoza et al., 2018*). Targeting such activities may be detrimental for cancer cell survival and present novel treatment strategies as exemplified by the synthetic lethality between BRCA1/2 mutated, homologous recombination (HR)-defective tumours, and inactivation of single strand break (SSB) repair by Poly (ADP-ribose) polymerase (PARP) inhibitors (*Ashworth, 2008*; *Bryant et al., 2005*; *Dobbelstein and Sørensen, 2015*). Regrettably, highly aggressive and genetically unstable cancers often rewire checkpoint/repair pathways that contribute to chemotherapy resistance and escape from targeted therapies (*Marzio et al., 2019*; *Yazinski et al., 2017*).

Uncoupling of polymerase and helicase activities during oncogene-induced RS leads to single-stranded DNA (ssDNA) bound by replication protein A (RPA) and activation of ATR for protection of stalled replication forks and/or repair of damaged DNA. ssDNA-RPA mediated recruitment of ATR (through ATRIP) promotes phosphorylation of H2AX and activation of CHK1. ATR-CHK1 then orchestrate the RS response through various downstream targets including phosphorylation and inactivation of phosphatase CDC25 and inhibition of CDK activity (*Kotsantis et al., 2018*). Hence, inhibition of ATR has been recognized as an attractive pharmacological strategy for amplification of RS in cancer cells (*Lecona and Fernandez-Capetillo, 2018*). Indeed, Toledo et al. demonstrated that unscheduled origin firing upon ATR inhibition caused replicative stress and eventually replication catastrophe due to RPA depletion (*Toledo et al., 2013*). However, activation of CHK1 was shown to circumvent replication catastrophe associated with ATR inhibition (*Buisson et al., 2015*) and co-inhibition of CHK1 produced synthetic lethality in cancer cells through CDK-dependent origin firing (*Sanjiv et al., 2016*). Accumulating evidence demonstrates significant crosstalk between different DNA damage response (DDR) kinases. For instance, DNA-PK was shown to facilitate ATR signalling at stalled replication forks (*Lin et al., 2018*) and reinforce ATR-mediated checkpoint activation through RPA phosphorylation (*Ashley et al., 2014*).

The WEE1 checkpoint kinase controls DNA replication and the G2/M transition through phosphorylation of CDKs (*Parker and Piwnica-Worms, 1992*; *Watanabe et al., 1995*). Inhibition of WEE1 removes the negative phosphorylation on Y15-CDK2 (in concert with CDC25) resulting in overactivation of CDK2, increased origin firing and reduced replication fork processivity. Indeed, deletion of CDK2 was shown to desensitize cancer cells to WEE1 inhibition (*Heijink et al., 2015*). WEE1 also phosphorylates and blocks CDK1 activity in response to DNA damage, activating the G2/M checkpoint to prevent unscheduled mitosis (*Russell and Nurse, 1987*). Consequently, WEE1 inhibition can force cells to undergo pre-mature mitosis and mitotic catastrophe and combined inhibition of WEE1 and ATR is synergistic in vitro and in vivo, thus representing a promising therapeutic approach (*Bukhari et al., 2019*; *Jin et al., 2018*; *Ruiz et al., 2016*; *Young et al., 2019*). However, the relative contribution of different checkpoint kinases and repair/replication factors in regulating recovery following treatment with these drugs remains to be elucidated. Overexpression of WEE1 is associated with poor patient outcome in several malignancies (*Iorns et al., 2009*; *Slipicevic et al., 2014*), and to date few markers predicting response to WEE1 inhibition have been identified.

In this study, we performed a systematic analysis of AZD1775 sensitivity in relation to global proteome and transcriptome in a panel of breast cancer cell lines and identified a group of basal-like breast cancer (BLBC) cell lines with low PTEN expression that were highly sensitive to AZD1775 monotherapy. Importantly, we show that PTEN-positive BLBC cell lines recovered following AZD1775 treatment through a DNA-PK-mediated mechanism and demonstrate potent synergy between AZD1775 and DNA-PK inhibitor NU7441. AZD1775 induced the phosphorylation of DNA-PKc (here referred to as DNA-PK) and CHK1 independently of ATR and deletion of DNA-PK prevented recovery of proliferation following AZD1775 monotherapy, establishing a novel synthetic lethal interaction between DNA-PK and AZD1775 in BLBC cells. Together, these results identify PTEN as a potential biomarker that may help to guide efficient WEE1 inhibitor therapy in the clinical setting, and show that a key function of PTEN and DNA-PK is to protect BLBC cells from lethal DNA damage by suppressing excessive replication stress induced by AZD1775.

## Results

### Differential response and recovery of basal-like breast cancer (BLBC) cell lines to AZD1775 monotherapy

To assess breast cancer cell line sensitivity to AZD1775, we performed a high-content imaging (HCI)-based analysis of 16 cell lines representing major subtypes of breast cancer treated with AZD1775 (*Figure 1A*). Changes in cell numbers following AZD1775 treatment were retrieved at single-cell resolution and visualized as dose response curves (*Figure 1A*, *Figure 1—figure supplement 1A*, *Supplementary file 1*). Inhibition of proliferation in response to AZD1775 was further validated by crystal violet staining assays (*Figure 1—figure supplement 1B* and data not shown). Among all the cell lines tested, the BLBC subtype displayed the highest sensitivity to AZD1775 acutely compared with luminal cell lines (*Figure 1B*, *Figure 1—figure supplement 1A*, *Supplementary file 1*), in

agreement with recent studies (*Bukhari et al., 2019*; *Jin et al., 2018*). To assess whether the response to AZD1775 persisted following removal of inhibitor, we evaluated cellular recovery by drug washout experiments. Since ATR inhibition by AZD6738 has been reported to be synergistic with AZD1775 (*Jin et al., 2018*), cellular recovery following single-agent treatment was compared to AZD1775 + AZD6738 combination treatment. Importantly, even though the combination was synergistic acutely in multiple BLBC cell lines, it provided no clear synergism in a subset of BLBC cell lines that were highly sensitive to AZD1775 monotherapy (*Figure 1C*, *Figure 1—figure supplement 1C*). Instead, AZD1775 mono-treatment prevented recovery of the highly sensitive BLBC cell lines as opposed to the group of BLBC cell lines that re-proliferated after AZD1775 monotherapy drug washout (hereafter referred to as sensitive and recovering BLBCs, respectively) (*Figure 1C,D and E*, *Figure 1—figure supplement 1D*). Consistent with previous reports (*Bukhari et al., 2019*; *Jin et al., 2018*), AZD1175 treatment was accompanied by decreased tyrosine 15 phosphorylation of CDK1 and CDK2 (i.e. CDK activation), while the acute synergistic effect of the combination treatment enhanced DNA damage as shown by an increased phosphorylation on S139-γH2AX (*Figure 1—figure supplement 1E and F*). The increased γH2AX phosphorylation in response to AZD1175 or the combination treatment was attenuated by the CDK2 inhibitor Milciclib (*Figure 1—figure supplement 1G*), confirming that the DNA-damaging effect of WEE1 inhibition is attributed to CDK activation.

To investigate the mechanism of recovery from drug treatment, we analysed replication-associated DNA damage by EdU/γH2AX staining in several BLBC cell lines. AZD1775 mono-treatment significantly increased the fraction of γH2AX in S-phase (EdU+) cells, particularly in AZD1775-sensitive BLBC cells (HCC38 and HCC1937) while recovering BLBC cells (BT20 and MDA-MB-231) showed minimal replication-associated DNA damage (γH2AX+/EdU+) in response to AZD1775 treatment (*Figures 1F* and *2F*). Assessment of apoptosis by AnnexinV/PI staining showed that AZD1775 monotherapy induced apoptotic cell death in sensitive HCC38 cells, while MDA-231 cells recovering following treatment displayed limited induction of apoptosis (*Figure 1—figure supplement 2*). Together, these results reveal differential sensitivity and outcomes of BLBC cells treated with AZD1775 and indicate that the AZD1775 + AZD6738 combination synergy recently reported for BLBC (*Jin et al., 2018*) only applies to a fraction of this breast cancer subtype.

## PTEN predicts sensitivity and response to AZD1775 monotherapy

While several potential response markers for DDR kinase inhibitors have been identified (*Brandsma et al., 2017*), biomarkers predicting AZD1775 response are limited. To identify markers associated with AZD1775 sensitivity in breast cancer, we performed a systematic analysis of AZD1775 response profiles in relation to the global proteome and transcriptome in the panel of breast cancer cell lines. The proteomics profiling identified proteins from 11408 genes with 9222 proteins with quantitative data across all sample replicates (*Figure 2—figure supplement 1A*). Analysis of the proteomics and transcriptomics data showed a significant correlation (Spearman's averages correlation was 0.63 and median 0.69) between protein and mRNA expression levels across the cell line models (*Figure 2—figure supplement 1B*, *Supplementary files 2* and *3*). Similarly, we found a general connection between DNA copy-number aberrations and changes to mRNA and protein expression levels (data not shown). Unsupervised hierarchical clustering of the breast cancer proteomics data using the intrinsic subtype classifier PAM50 genes showed sample clusters separating luminal from BLBC cell lines (*Figure 2—figure supplement 1C*). On the gene level we found clusters of functionally linked genes relating to the main transcriptional features of breast cancer (*Figure 2—figure supplement 1D*; *Neve et al., 2006*). Correlation of AZD1775 drug sensitivity to all proteins in the cell line panel identified PTEN expression as one of the strongest predictors of sensitivity to AZD1775 both at the protein (p=0.0002) and RNA levels (p=0.0002) (*Figure 2A and B*, *Figure 2—figure supplement 1E and F*). Expression levels of other proteins associated with AZD1775 sensitivity (TP53I11, CTPS1 and CDC7, *Figure 2A*) did not discriminate between sensitivity and recovery of BLBC cells post AZD1775 treatment and was mainly linked to molecular subtype (basal-like vs luminal cells) (*Figure 2—figure supplement 2A*). The negative correlation between PTEN expression and AZD1775 response was validated in several breast cancer cell lines by immunoblotting and drug response profiling (*Figure 2—figure supplement 2B*). To further explore the relationship between PTEN expression and dependency on WEE1 in breast cancer cell lines, we assessed WEE1 RNAi loss-of-function gene effects using the depmap portal (*McFarland et al.,*

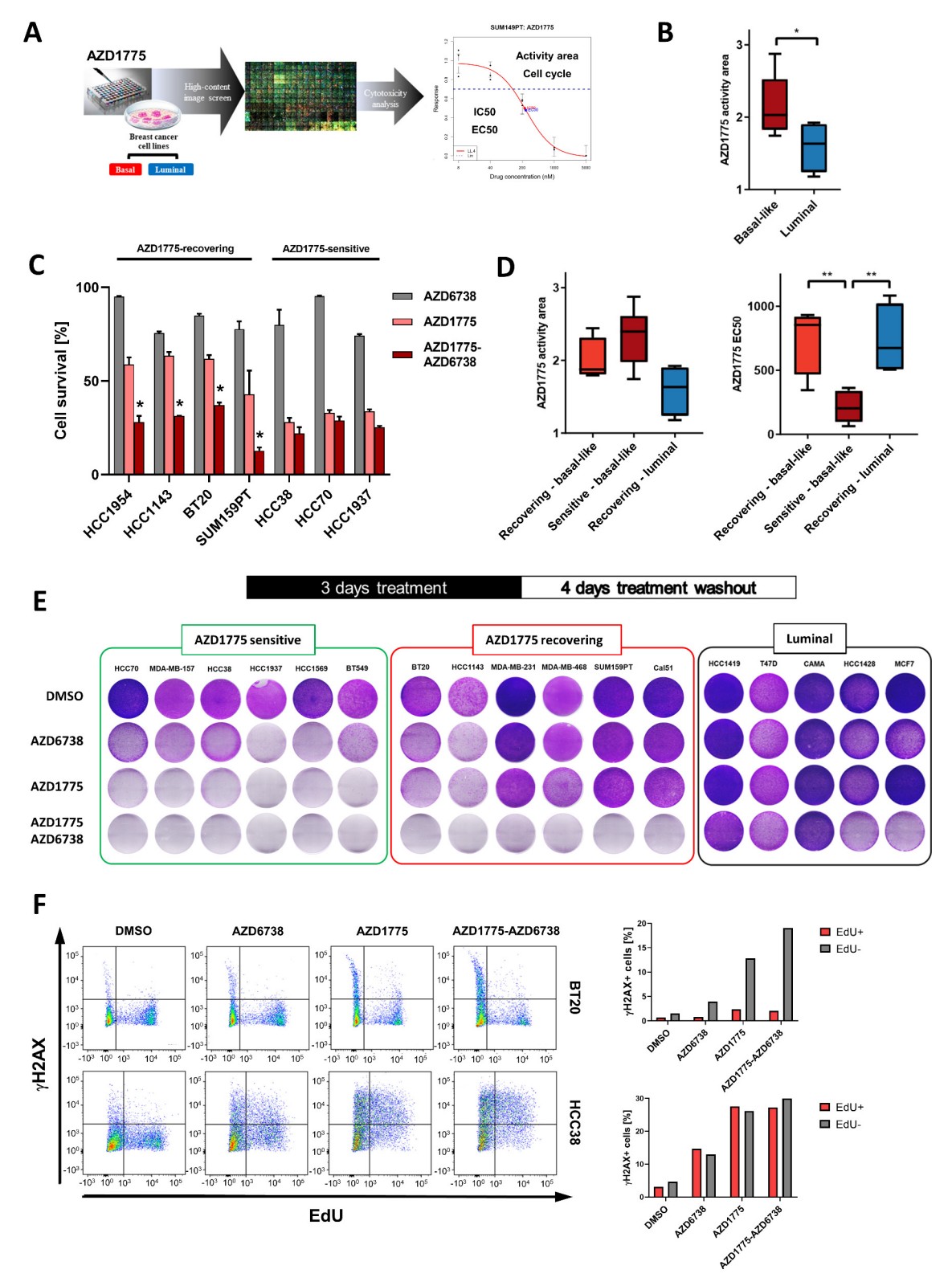

**Figure 1.** Differential response and recovery of basal-like breast cancer (BLBC) cell lines to AZD1775 monotherapy. (**A**) Overview of the HCI AZD1775 screen in breast cancer cell lines and drug response characteristics. Activity Area - AA, EC50 - concentration of compound achieving 50% of the maximum effect, IC50 - concentration of compound achieving 50% reduction in cell number, and cell cycle profile. AZD1775 response profile of SUM149PT cells is shown (see also *Figure 1—figure supplement 1A*, *Figure 6—figure supplement 1G* and *Supplementary file 1*). (**B**) Comparison of

*Figure 1 continued*

acute AZD1775 sensitivity between basal-like and luminal breast cancer cell lines measured by HCI analysis. The AZD1775 response characteristics (activity area) depicted correspond to the area above the response curve. Boxes are coloured according to AZD1775 response in basal-like and luminal BC cell lines, respectively. * denotes p<0.05 as determined using Student's t-test. Data are represented as mean ± SD. (C) Acute response to AZD1775, AZD6738 or the combination relative to DMSO-treated control in different BC cell lines. Cell numbers were analysed by crystal violet staining and quantified by colorimetry after 72 hr treatment. Data shown refer to a single concentration of AZD1775 (500 nM), AZD6738 (1 µM) or their combination (fixed ratio, 1:2). Error bars indicate standard deviation calculated from three independent experiments. Drug combination effects were calculated, based on the ratio and concentrations above, using the CImbinator online tool (*Flobak et al., 2017*). Both AZD1775-sensitive (CI >1.0) and AZD1775-recovering (CI <1.0) BC cell lines are depicted and drug synergy is marked *. The following CI values were calculated; HCC1954 = 0.31; HCC1143 = 0.18; BT20 = 0.45; SUM159PT = 0.34; HCC38 = 2.06; HCC70 = 1.09; HCC1937 = 1.57. (D) Comparison of acute AZD1775 response between recovering basal-like, sensitive basal-like and luminal breast cancer cell lines (recovering) measured by HCI analysis. The AZD1775 response characteristics were set to the area above the response curve (Activity area) *left*, and EC50, *right*. Boxes are coloured according to AZD1775 response of the different BC cell lines (recovering-basal, sensitive-basal and recovering-luminal). ** denotes p<0.01 as determined using Student's t-test. (E) Recovery of proliferation following removal of AZD1775 (500 nM), AZD6738 (1 µM) or their combination was analysed by crystal violet staining and quantified by colorimetry. BC cell lines were treated for three days and allowed to recover for an additional four days without the drugs. Representative images of each cell line (from >three independent experiments) treated with indicated drugs and stained with crystal violet are shown. (F) Representative flow cytometry dot plots (left panels, two independent experiments) showing the distribution of EdU and γH2AX incorporation in recovering BT20, and sensitive HCC38 cells, following 24 hr treatment with AZD1775 (500 nM), AZD6738 (1 µM) or their combination as compared to DMSO control. Labelling of γH2AX-positive cells in the replicating fraction (EdU-positive - *red*) and γH2AX-positive cells in the non-replicating fraction (EdU-negative - *gray*) of each cell line is shown (right panels).

The online version of this article includes the following source data and figure supplement(s) for figure 1:

**Source data 1.** Differential response and recovery of basal-like breast cancer (BLBC) cell lines to AZD1775 monotherapy.
**Figure supplement 1.** Differential response and recovery of basal-like breast cancer (BLBC) cell lines to AZD1775 monotherapy.
**Figure supplement 2.** Differential response and recovery of basal-like breast cancer (BLBC) cell lines to AZD1775 monotherapy.

*2018*; *Tsherniak et al., 2017*) and estimated the reliance on PTEN protein expression for viability. Strikingly, low PTEN protein was significantly correlated with a greater dependency on WEE1 in our proteomic dataset (Pearson's correlation, r = 0.70, p=0.008) (*Figure 2C*). Co-dependency between WEE1 and PTEN protein was also found in an independent proteome dataset generated in the Gygi lab (*Nusinow et al., 2020*) (Pearson's correlation, r = 0.40, p=0.041) (*Figure 2—figure supplement 2C*). To investigate if PTEN regulates recovery following AZD1775 monotherapy, we first generated two isogenic PTEN knockout lines of the recovering BLBC cell line MDA-MB-231 using CRISPR-Cas9 genome editing (*Figure 2D*). PTEN-proficient and PTEN-deficient MDA-MB-231 cells were treated with AZD1775 and cellular recovery was assessed after removal of AZD1775. Consistently, deletion of PTEN significantly attenuated recovery of MDA-MB-231 cells (*Figure 2E*). We next assessed replication-associated DNA damage in PTEN-proficient and PTEN-deficient MDA-MB-231 cells and in AZD1775 sensitive HCC1937 cells by high-content immunofluorescence microscopy of γH2AX/EdU. As shown in *Figure 2F* and *Figure 2—figure supplement 2D*, AZD1775 treatment significantly increased phosphorylation of γH2AX in replicating (EdU+) PTEN-deficient cells as compared to PTEN-proficient cells, supporting an impaired replication checkpoint and increased sensitivity to AZD1775-induced RS in the PTEN-deleted cells. Accordingly, re-expression of PTEN in hypersensitive PTEN-negative HCC1937 cells significantly reduced AZD1775 sensitivity as compared to HCC1937 empty vector (EV) control cells (*Figure 2G*). Similar results were obtained using an independent WEE1 inhibitor (PD0166285) (*Figure 2—figure supplement 2E*). Knockdown of WEE1 using siRNAs also reduced the viability of PTEN-deficient as compared to PTEN-proficient cells (*Figure 2—figure supplement 3*). Finally, colony formation assays confirmed improved ability of PTEN-reconstituted HCC1937 cells to reproliferate post AZD1775 monotherapy (*Figure 2H*). In conclusion, PTEN can not only be used as a new biomarker for AZD1775 sensitivity, but apparently also plays an active role in processes that have an impact on recovery from AZD1775 treatment.

## ATR inhibition by AZD6738 exacerbates AZD1775-induced RS and abrogates recovery of replication

We next looked closer into the subgroup of BLBC cell lines recovering post AZD1775 monotherapy. In agreement with the results in *Figure 1* and with recent data published during the preparation of

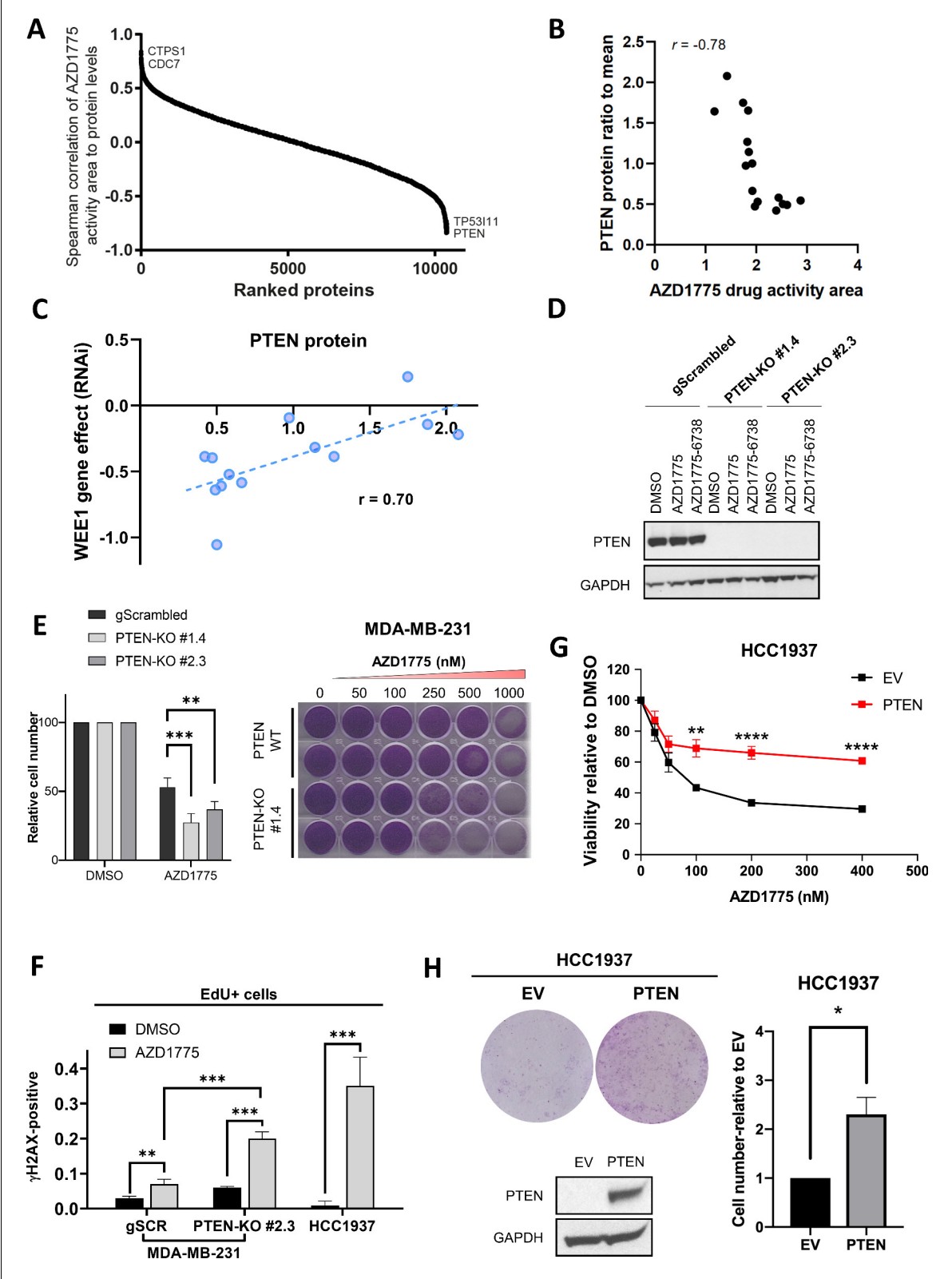

**Figure 2.** PTEN predicts sensitivity and response to AZD1775 monotherapy. (**A**) Association of AZD1775 drug response to protein levels. AZD1775 drug response activity area was associated with proteins levels from the 16 cell lines in the screen by Spearman correlation. (**B**) Correlation between PTEN protein (ratio to mean) and response to AZD1775 (drug response area). Pearson correlation coefficient r = −0.78, p=0.0002. (**C**) Correlation analysis of WEE1 RNAi gene dependency (combined RNAi, DEMETER2 model, depmap portal [*McFarland et al., 2018*; *Tsherniak et al., 2017*]) and

*Figure 2 continued on next page*

*Figure 2 continued*

PTEN protein levels (ratio to mean) in 13 breast cancer cell lines (Pearson's correlation, r = 0.70, p=0.008; Spearman's correlation, r = 0.70, p=0.01). (D) MDA-MB-231 gScrambled and PTEN knockout (KO) clones (#1.4 and #2.3) were treated for 24 hr with AZD1775 (500 nM) or combination of AZD1775 and AZD6738 (1 μM), and whole cell lysates were immunoblotted and probed with the indicated antibodies. (E) Quantification of recovery of proliferation after 72 hr treatment with increasing concentrations of AZD1775 in the indicated isogenic cell lines (*left*). Data shown are mean ± SD of three independent experiments, **p=0.003 and ***p=0.0003, assessed by Student's t-test. Images of recovery of proliferation of MDA-MB-231 scrambled control (PTEN-WT) and PTEN-KO (#1.4) cells treated with indicated concentrations of AZD1775 and assessed by staining with crystal violet (*right*). (F) Quantification of DNA damage by HCI analysis of γH2AX-positive cells in the replicating, EdU+ fraction. PTEN-proficient (MDA-MB-231 scrambled control), PTEN-deficient (PTEN-KO #2.3) and PTEN-deleted HCC1937 cells were treated with AZD1775 (500 nM) or DMSO for 24 hr. Graphs show the proportions of EdU/γH2AX double-positive cells, ** indicates p=0.098 and ***p<0.001 as assessed by Student's t-test. (G) Quantification of AZD1775 response in HCC1937 (EV, PTEN-negative) cells and HCC1937 cells with reconstituted PTEN (PTEN-positive). Cell viability was analysed by alarmarBlue assay (*upper panel*). Error bar indicates SEM calculated from five replicates. **p<0.002 and ****p<0.0001 assessed by Student's t-test. (H) Recovery of proliferation (10 days) of EV and PTEN restored HCC1937 cells following 72 hr treatment with AZD1775 (100 nM) was analysed by crystal violet staining (representative image is shown, *top left panel*). Recovery of proliferation was measured by crystal violet staining. Error bar indicates SEM calculated from two independent experiments. * indicates p=0.003 assessed by Student's t-test (*right panel*). PTEN protein expression was analysed by western blotting and GAPDH level served as loading control (*lower left panel*).

The online version of this article includes the following source data and figure supplement(s) for figure 2:

**Source data 1.** PTEN predicts sensitivity and response to AZD1775 monotherapy.
**Figure supplement 1.** PTEN predicts sensitivity and response to AZD1775 monotherapy.
**Figure supplement 2.** PTEN predicts sensitivity and response to AZD1775 monotherapy.
**Figure supplement 3.** PTEN predicts sensitivity and response to AZD1775 monotherapy.

the manuscript (*Bukhari et al., 2019*; *Jin et al., 2018*), combination treatment using AZD1775 administrated concurrently with AZD6738 efficiently inhibited recovery of proliferation of MDA-MB-231 cells following drug-washout, while the mono-treatments were essentially ineffective (*Figure 3A*). Supporting these results, AZD1775 or AZD6738 monotherapy did not affect tumour growth compared with vehicle, while the combination of AZD1775 + AZD6738 clearly mitigated tumour growth in vivo (*Figure 3B*). We next monitored progression through the cell cycle of MDA-MB-231 cells treated with AZD1775, AZD6738, or the combination. After one hour of drug pretreatment, the cells were pulse-labelled with EdU and released for different times up to 24 hr post-treatment. As shown in *Figure 3—figure supplement 1A*, cells treated with the combination stalled in S-phase and failed to proceed through the cell cycle, while a significant proportion of cells treated with the single drug divided and re-entered G1-phase. Consequently, EdU labelling after 23 hr of drug treatments showed that only 13% of cells treated with the AZD1775 + AZD6738 combination incorporated EdU, while a substantial fraction of mono- and vehicle-treated cells were in active replication (*Figure 3C*). These data were confirmed in vivo using a xenograft model with MDA-MB-231 cells where the combination treatment strongly reduced BrdU incorporation, whereas the mono-therapies had only marginal effects (*Figure 3—figure supplement 1B*). Consistent with this, removal of AZD1775 (or AZD6738) restored replication in MDA-MB-231 and BT20 cells in vitro (*Figure 3—figure supplement 1C*), recapitulating the recovery phenotype and demonstrating that AZD1775-induced RS is well tolerated in these BLBC cells (*Figures 1E* and *3A* and B). By contrast, the AZD1775 + AZD6738 combination completely prevented EdU incorporation and recovery of proliferation even at low doses of AZD1775 supporting a wide therapeutic window and high potency of AZD1775 when combined with AZD6738 (*Figure 3D* and *Figure 3—figure supplement 1C*). Further, AZD1775 + AZD6738 treatment strongly increased the number of β-gal positive cells, suggesting that the combination leads to cellular senescence (*Figure 3—figure supplement 1D*), a state of persistent cell cycle arrest (*d'Adda di Fagagna, 2008*; *Halazonetis et al., 2008*).

To investigate how the mono- and combo treatments affected replication stress-induced DNA damage, we assessed formation of single-stranded DNA (ssDNA) as an indication of stalled replication forks, measured by immunofluorescence of native BrdU by FACS. In addition, we analysed induction of DNA damage by γH2AX staining as well as expression of RAD51. Both AZD1775 mono-treatment and the AZD1775 + AZD6738 combination increased formation of ssDNA, the combination with very rapid kinetics (*Figure 3E* and *Figure 3—figure supplement 2A*). Although the single-agent treatments induced replication-associated DNA damage, the number of γH2AX foci was strongly reduced in the mono-treated cells after 3 days of drug wash-out, indicative of ongoing DNA repair, while a high proportion of the combo-treated cells retained γH2AX foci

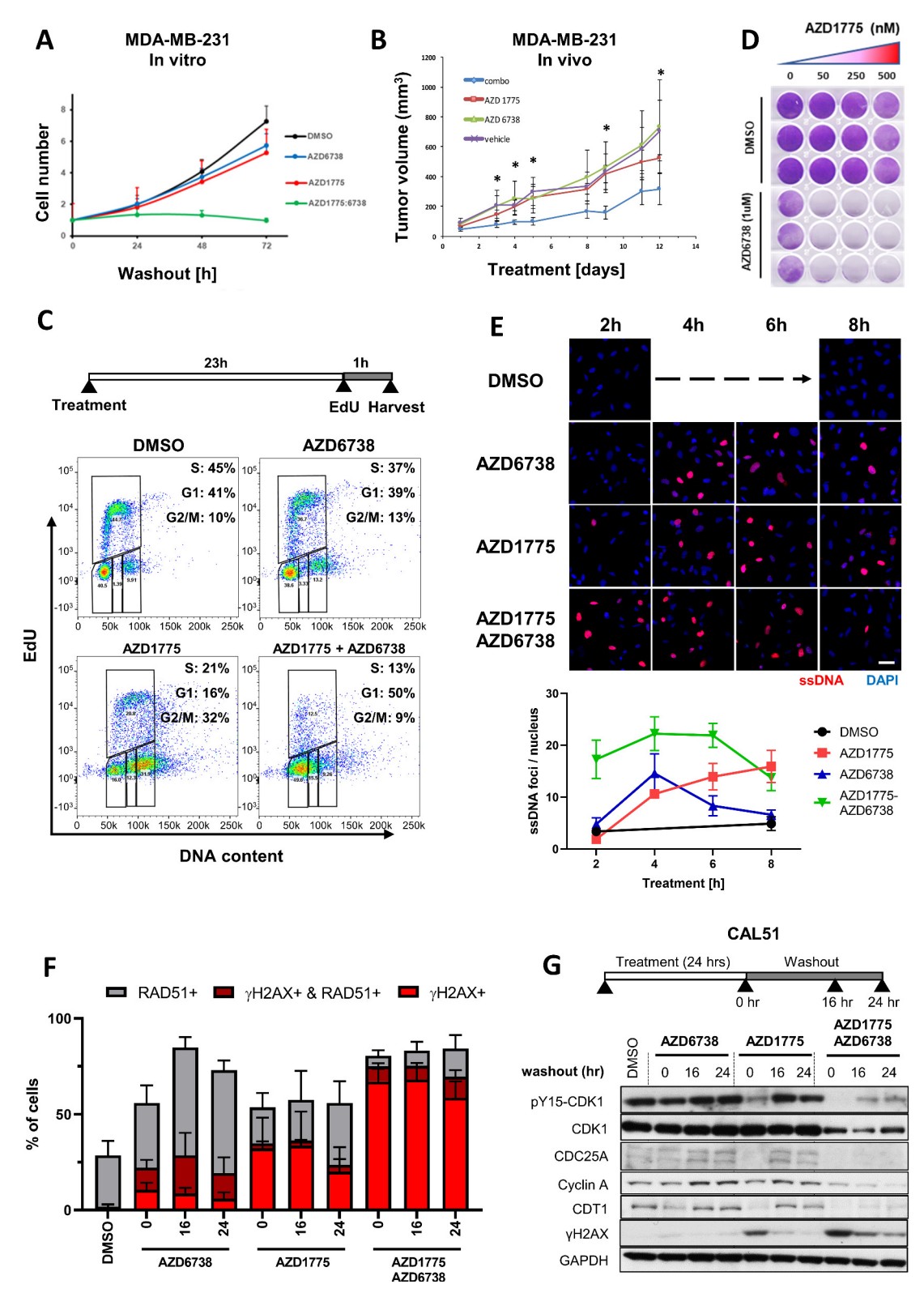

**Figure 3.** ATR inhibition by AZD6738 exacerbates AZD1775-induced RS and abrogates recovery of replication. (A) Quantification of cell proliferation post-drug treatment as indicated in MDA-MB-231 cells. Error bars indicate standard deviation calculated from three independent experiments. Cell proliferation was determined by trypan blue cell counting in triplicates for each condition. (B) Effect of the combination treatment with AZD1775 and AZD6738 on tumour growth in vivo. Mice (NOD/SCID female mice, 6–8 weeks old) were injected orthotopically with $10^6$ MDA-MB-231 cells. As tumours

*Figure 3 continued on next page*

*Figure 3 continued*

reached 67.92mm3 +/- 10.66 mm3 (mean +/- SEM) mice started receiving one of the following treatments by oral gavage: vehicle (n = 4), AZD6738 25 mg/kg (n = 4), AZD1775 25 mg/kg (n = 4) or AZD6738 25 mg/kg + AZD1775 25 mg/kg ('combo') (n = 6) for 12 days. Tumours were measured periodically, and tumour volume calculated with the formula: $V = D \times d2 \times \pi/6$, where V is volume, D is the largest diameter, d is the shortest diameter. *p<0.05 as assessed by Student's t-test with respect to combination treatment vs. vehicle. The statistical analysis is based on tumour growth index that is the relative increase in tumour volume at a particular time point compared with the tumour volume at the start of treatment for each tumour. (C) DNA replication in MDA-MB-231 cells treated as indicated was assessed by flow cytometry. Cells were pulse-labelled with EdU for one hour at the end of the treatment before fixation and staining. Cells were gated for EdU and DNA content (propidium iodide) to determine cell cycle phase. Y- and X-axes represent EdU incorporation and DNA content, respectively. (D) MDA-MB-231 cells were treated with increasing concentrations of AZD1775 alone (50, 250 and 500 nM) or in combination with AZD6738 (1 µM) for three days, and recovery of proliferation was assessed four days post-treatment by crystal violet staining. (E) Representative, high-magnification images related to *Figure 3—figure supplement 2A* visualizing ssDNA DNA by native BrdU staining (*red*) in response to AZD1775, AZD6738 or the combination treatment in MDA-MB-231 cells as indicated. Quantification of ssDNA in response to the different treatment times is depicted with error bars indicating SEM calculated from five replicates. Scale bar = 30 µM. (F) Quantification of data from HCI analysis of MDA-MB-231 cells treated with AZD1775 (500 nM), AZD6738 (1 µM) or their combination as indicated (*Figure 3—figure supplement 2C*). The staining signals of RAD51 and γH2AX were measured at single cell resolution (DAPI-stained area) and the fraction of RAD51/γH2AX-positive cells were quantified and shown as histograms with error bars, standard deviation derived from three independent replicates. (G) Expression level and phosphorylation of cycle regulatory proteins during the course of recovery following treatment with AZD1775 (500 nM), AZD6738 (1 µM), or their combination in recovering Cal51 cells. CDC25A, cyclin A, CDT1, phosphorylation of CDK1-Y15 and γH2AX was analysed by western blotting as indicated. GAPDH expression level served as loading control.

The online version of this article includes the following source data and figure supplement(s) for figure 3:

**Source data 1.** ATR inhibition by AZD6738 exacerbates AZD1775-induced RS and abrogates recovery of replication.
**Figure supplement 1.** ATR inhibition by AZD6738 exacerbates AZD1775-induced RS and abrogates recovery of replication.
**Figure supplement 2.** ATR inhibition by AZD6738 exacerbates AZD1775-induced RS and abrogates recovery of replication.

(*Figure 3F* and *Figure 3—figure supplement 2B and C*). Co-staining of γH2AX and RAD51 showed that the AZD1775 + AZD6738 combination also reduced RAD51 expression compared to cells treated with the single drugs, which retained or even increased RAD51 expression (*Figure 3F* and *Figure 3—figure supplement 2C*). To test more directly how these drugs influence expression of replication regulatory proteins after treatment washout, we analysed CDK activator phosphatase CDC25, replication licensing factor CDT1, and Cyclin A. Recovering BLBC cell lines (Cal51 and MDA-MB-231) were treated with either AZD1775, AZD6738 or their combination and subsequently released into drug-free media. Substantiating the post-treatment recovery results, CDC25, CDT1 and Cyclin A was re-expressed following removal of the single drug treatments, while the combination treatment severely impeded re-expression of these proteins (*Figure 3G* and *Figure 3—figure supplement 2D*). Together, these data demonstrate that the AZD1775 + AZD6738 combination permanently abrogates replication and proliferation in the subgroup of recovering BLBC cell lines.

## DNA-PK is phosphorylated in response to AZD1775 and preserves CHK1 phosphorylation independent of ATR

In an attempt to identify DDR factors in addition to ATR that might regulate sensitivity to AZD1775, we initiated a high-content image-based RNA interference (RNAi) screen of 300 DDR genes in MDA-MB-231 cells. siRNAs targeting several DDR factors with well-established functions in regulation of RS and HR repair, such as RPA1 and BRCA1, increased sensitivity to AZD1775 as expected (*Figure 4A* and *Figure 4—figure supplement 1A*). Gene ontology (GO) enrichment analysis showed that depletion of HR-associated genes significantly reduced viability in response to AZD1775 treatment (p=0.035) (*Figure 4B*), consistent with a previous study demonstrating synthetic lethality between WEE1 inhibition and defective HR pathway (*Aarts et al., 2015*). Interestingly, siRNAs targeting non-homologous end joining (NHEJ) factors including DNA-PK (*PRKDC*) also decreased viability of MDA-MB-231 cells treated with AZD1775 (*Figure 4A and B*, *Figure 4—figure supplement 1A*), and concurrent treatment with DNA-PK inhibitors (NU7441) and AZD1775 synergistically reduced cell viability in multiple BLBC cell lines (*Figure 4C* and *Figure 4—figure supplement 1B*). We therefore decided to take a closer look at DNA-PK regulation and function in response to AZD1775 treatment in BLBC cells, in particular in relation to induction of replication stress, DNA damage and CHK1 activation.

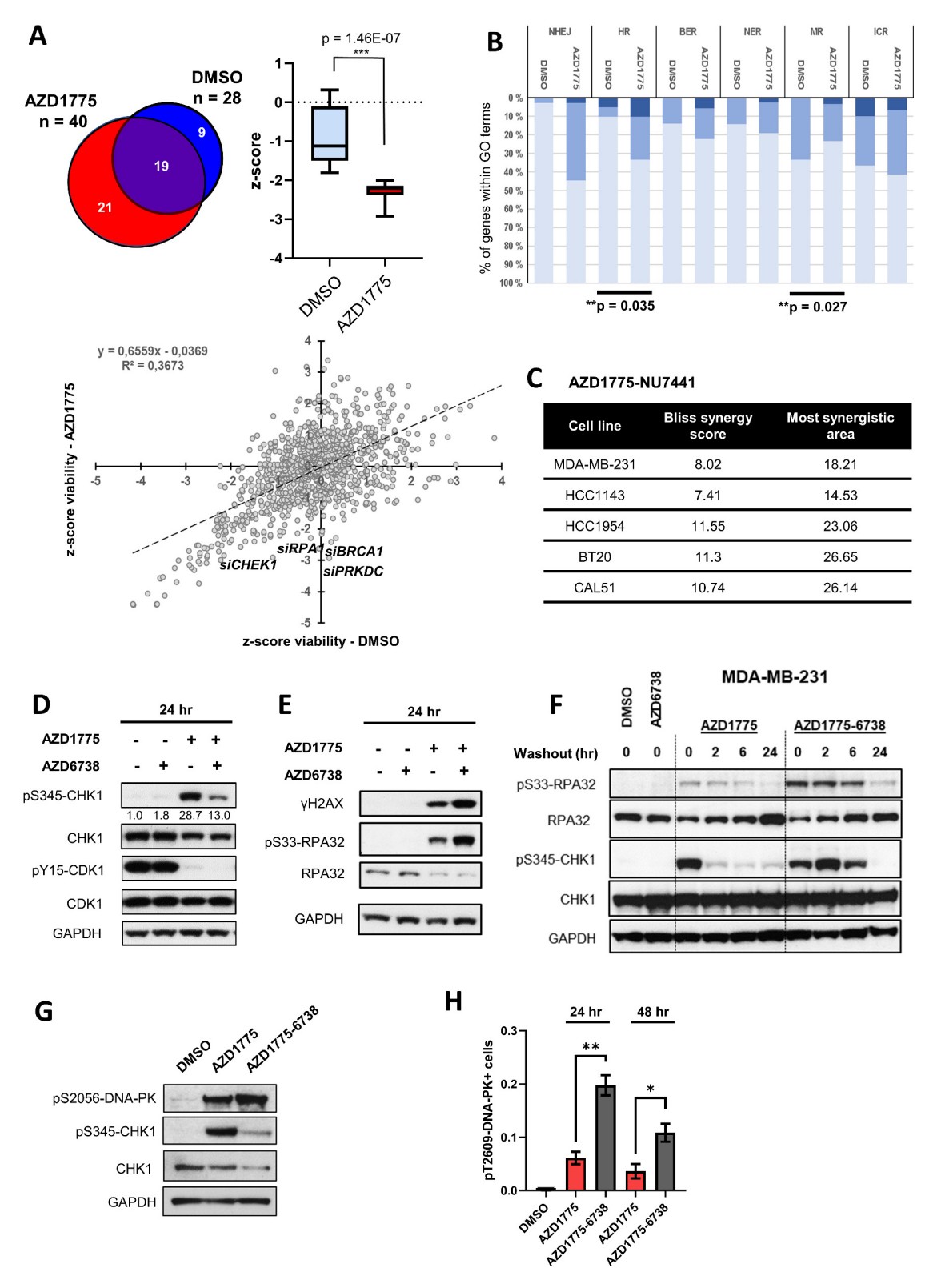

**Figure 4.** DNA-PK is phosphorylated in response to AZD1775 and preserves CHK1 phosphorylation independent of ATR. (**A**) MDA-MB-231 cells were transfected with a library of siRNAs targeting 300 DNA repair genes and analysed for siRNA hits (three siRNAs per gene) reducing viability in response to AZD1775 treatment by high-content image analysis. siRNAs targeting 28 genes resulted in significant loss of viability in control conditions compared to 40 genes in combination with AZD1775. 21 genes selectively reduced viability when combined with AZD1775 (p=1.46E-7) (*Upper panel*). Scatter plot

*Figure 4 continued on next page*

Figure 4 continued

visualisation of the viability z-score correlation of all siRNAs in the screen. Target genes (siRNAs) affecting MDA-MB-231 cell viability from the DDR siRNA library in response to 72 hr treatment of AZD1775 are marked in the plot (*Lower panel*). (B) Grouping of different DNA repair pathway-associated genes from high-content siRNA screen based on gene ontologies. Knockdown of homologous recombination (HR) genes associates with increased sensitivity to AZD1775 (p=0.035). Bars are coloured according to z-score. z-score < −2 (dark), z-score < −1 (medium) and z-score - n.s. (light). (C) BLBC cell lines were treated with NU7441 and AZD1775 at different concentrations for three days. Drug synergy scores (Bliss model) calculated from dose-response matrices are shown as indicated. (D) MDA-MB-231 cells were treated for 24 hr with the indicated drugs or DMSO control as specified. Whole cell lysates were collected, and expression and phosphorylation of CHK1 and CDK1 was analysed by immunoblotting with the specified antibodies. Expression of phosphorylated CHK1 was quantified by normalizing band density to total CHK1 band density. GAPDH level was used as loading control. (E) MDA-MB-231 cells were treated as in D and protein expression and phosphorylation analysed with the specified antibodies, as depicted. (F) MDA-MB-231 cells were treated as in D and harvested or allowed to recover in the absence of the compounds for 2, 6 or 24 hr before harvesting. Total cell lysates were immunoblotted with the indicated antibodies. (G) MDA-MB-231 cells treated for 24 hr with the indicated drugs and phosphorylation of DNA-PK (pS2056) and CHK1 (pS345) was assessed by immunoblotting. GAPDH level was used as loading control. (H) MDA-MB-231 cells were treated with the indicated drugs for 24 or 48 hr and phosphorylation of DNA-PK (pT2609) was analysed by high-content immunofluorescence microscopy. The proportion of pT2609-DNAPK-labelled cells represents mean of three biological replicates ± SEM. * and ** indicate p=0.029 and p=0.0036 respectively, as assessed by Student's t-test.

The online version of this article includes the following source data and figure supplement(s) for figure 4:

**Source data 1.** DNA-PK is phosphorylated in response to AZD1775 and preserves CHK1 phosphorylation independent of ATR.
**Figure supplement 1.** DNA-PK is phosphorylated in response to AZD1775 and preserves CHK1 phosphorylation independent of ATR.
**Figure supplement 2.** DNA-PK is phosphorylated in response to AZD1775 and preserves CHK1 phosphorylation independent of ATR.

Since CHK1 is a key downstream ATR effector substrate that is activated in response to ssDNA and DNA damage (*Kotsantis et al., 2018*; *Toledo et al., 2011*), we first assessed S345-CHK1 and S33-RPA phosphorylation in response to either AZD1775 mono-treatment or AZD1775 + AZD6738 combination treatment. Although AZD6738 acutely attenuated AZD1775-induced phosphorylation of S345-CHK1, phosphorylated CHK1 was preserved in cells treated with the AZD1775 + AZD6738 combination relative to total CHK1 levels (*Figure 4D*), and phosphorylation of RPA accumulated with the AZD1775 + AZD6738 combination (*Figure 4E*), suggesting activation of a compensatory pathway that preserves CHK1 phosphorylation and presumably replication arrest when ATR is inhibited. Further, CHK1 phosphorylation was sustained following wash-out of the two combined drugs as compared to wash-out after AZD1775 mono-treatment (*Figure 4F*). Previous studies have reported that DNA-PK maintains the stability of CHK1 in response to replication stress and DNA-PK-mediated phosphorylation of CHK1 was shown to inhibit origin firing when ATR is inactivated (*Buisson et al., 2015*). Indeed, we found that DNA-PK was phosphorylated by the combination treatment and interestingly, also by AZD1775 mono-treatment while total DNA-PK protein level was unaffected (*Figure 4G and H*, *Figure 4—figure supplement 1C and D*). Phosphorylated DNA-PK co-localized with pS345-CHK1 and ssDNA (*Figure 4—figure supplements 1E* and *2A*) and DNA-PK inhibitors (but not ATM inhibitors) attenuated phosphorylation of S33-RPA (*Figure 4—figure supplement 2B*). Finally, immunohistochemical staining of orthotopic xenograft tumours confirmed that DNA-PK is phosphorylated and activated in response to AZD1775 and AZD1775 + AZD6738 combination treatment in vivo (*Figure 4—figure supplement 2C*). Thus, AZD1775 induces phosphorylation of DNA-PK, potentially facilitating replication checkpoint arrest independent of ATR activity.

## DNA-PK regulates recovery of replication and survival in response to AZD1775

To directly test the function of DNA-PK in regulating cellular recovery following AZD1775 treatment, we knocked out DNA-PK in MDA-MB-231 cells using CRISPR-Cas9 genome editing. CRISPR gRNAs targeting RAD51 and PALB2 were used as controls for HR deficiency and recovery was assessed following AZD1775 mono-treatment as outlined in *Figure 5—figure supplement 1A*. Depletion of RAD51 and PALB2 attenuated recovery of proliferation, recapitulating the synergistic effects of the AZD1775 + AZD6738 combination (*Figure 5—figure supplement 1B*). Although we were unable to produce stable RAD51 and PALB2 knockout cell lines, we successfully generated two stable DNA-PK-negative cell lines. Knockout of DNA-PK was validated at the protein level by immunoblotting and immunofluorescence, and genetically by DNA sequencing (*Figure 5A* and *Figure 5—figure*

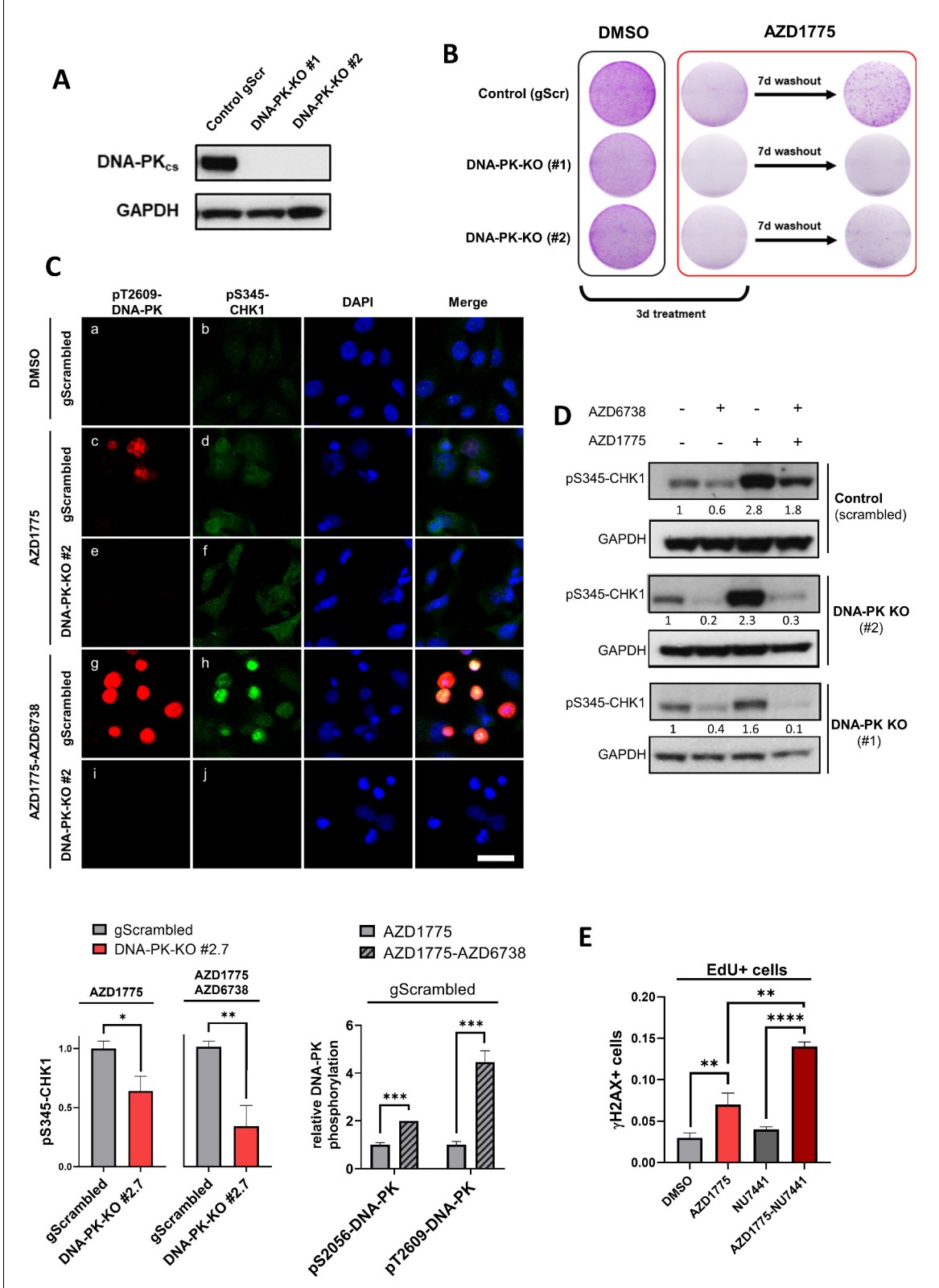

**Figure 5.** DNA-PK regulates recovery of replication and survival in response to AZD1775. (**A**) Immunoblot analysis of DNA-PK expression in MDA-MB-231 scrambled control and isogenic DNA-PK knock-out cell lines (clones #1 and #2). GAPDH expression was used as loading control. (**B**) Recovery of proliferation following AZD1775 mono-treatment (500 nM) in DNA-PK-proficient and DNA-PK-deficient MDA-MB-231 cells. Cells were treated for three days and allowed to recover for seven days without the drug and subsequently analysed by crystal violet staining. Representative images from >three

*Figure 5 continued on next page*

Figure 5 continued

independent experiments are shown. (C) *Top panel*: Immunofluorescence staining of pS345-CHK1 (*green*) and pT2609-DNA-PK (*red*) in DNA-PK-deficient (clone #2) and DNA-PK-proficient (control) MDA-MB-231 cells treated with 500 nM AZD1775, a combination of AZD1775 (500 nM) and AZD6738 (1 μM) or DMSO control. Scale bar = 30 μm. *Bottom panels*: Quantification of HCI-based analysis of pS345-CHK1 in DNA-PK-deficient (clone #2) and DNA-PK-proficient (control) MDA-MB-231 cells treated as above (*left*) and pT2609-DNA-PK as well as pS2056-DNA-PK in control cells (*right*). *p=0.01, **p=0.0032 and ***p<0.0003 as assessed by Student's t-test. (D) Immunoblot analysis of CHK1 phosphorylation (pS345-CHK1) in MDA-MB-231 scrambled control and DNA-PK knock-out cell lines (clones #1 and #2) treated with the indicated drugs for 24 hr. GAPDH expression was used as loading control. Expression of phosphorylated CHK1 was quantified by normalizing band density to GAPDH band density. (E) Quantification of DNA damage by HCI analysis of γH2AX-positive cells in the replicating (EdU+) fraction. MDA-MB-231 cells were treated with the indicated inhibitors for 24 hr. Graphs show the proportions of EdU/γH2AX double-positive cells, **p<0.01 and ****p<0.0001 as assessed by Student's t-test.

The online version of this article includes the following source data and figure supplement(s) for figure 5:

**Source data 1.** DNA-PK regulates recovery of replication and survival in response to AZD1775.
**Figure supplement 1.** DNA-PK regulates recovery of replication and survival in response to AZD1775.
**Figure supplement 2.** DNA-PK regulates recovery of replication and survival in response to AZD1775.

supplement 2A and B). DNA-PK-deficient cells were more sensitive to the DNA topoisomerase II inhibitor etoposide and proliferated at *slightly slower* rates compared to wild-type *cells* under normal culture conditions (*Figure 5—figure supplement 2C* and data not shown), consistent with previous reports (*Munck et al., 2012*). Importantly, recovery assays after drug wash-out showed that DNA-PK-deficient cells were unable to proliferate following AZD1775 mono-treatment, demonstrating that DNA-PK is critical for cellular recovery in response to AZD1775-induced RS in MDA-MB-231 cells (*Figure 5B*). To delineate the dependency on DNA-PK for DNA replication post AZD1775 treatment, we first analysed the phosphorylation status of DNA-PK and CHK1 in response to AZD1775 or the AZD1775 + AZD6738 combination by immunofluorescence analysis (*Figure 5C*). AZD1775 + AZD6738 combination increased phosphorylation of DNA-PK compared to AZD1775 single-agent treatment in the DNA-PK-proficient cells, presumably compensating for the inactivation of ATR by AZD6738 (*Figure 5C*, *image c vs. image g*). Phosphorylation of CHK1 (pS345-CHK1) was induced regardless of the presence or the absence of DNA-PK (*Figure 5C*, *image d vs. image f*). This result suggests that ATR compensates for the lack of DNA-PK and confirms that ATR is the primary kinase responsible for pS345-CHK1 in response to RS. However, while pS345-CHK1 was sustained in DNA-PK-proficient cells treated with the AZD1775 + AZD6738 combination (*Figure 5C*, *image h*), phosphorylation of CHK1 was reduced in DNA-PK-deficient cells (*Figure 5C*, *image j vs. h*), providing supporting evidence that DNA-PK-mediates phosphorylation of CHK1 when ATR is inhibited. Consistently, these results were confirmed by immunoblot analysis with pS345-CHK1 antibodies (*Figure 5D*), thus supporting a dual function of DNA-PK in promoting phosphorylation of CHK1 in the absence of ATR (AZD1775 + AZD6738 treatment) and regulating recovery of proliferation in the presence of ATR (AZD1775 treatment) (*Figure 5B,C and D*). To investigate recovery in more detail and its dependency on DNA-PK, we analysed replication-associated DNA damage (EdU/γH2AX) by high-content immunofluorescence microscopy. As shown in *Figure 5E* and *Figure 5—figure supplement 2D*, combined treatment with DNA-PK inhibitors (NU7441) and AZD1775 significantly increased DNA damage in the replicating fraction of MDA-MB-231 cells, compared to mono-treatment with either AZD1775 or NU7441. EdU/γH2AX staining by FACS analysis corroborated these results, showing a clear increase of γH2AX-labelled cells in the replicating fraction (EdU+) of DNA-PK-deficient compared to DNA-PK-proficient cells treated with AZD1775 (*Figure 5—figure supplement 2E*). Taken together, these data support a direct function of DNA-PK in facilitating replication arrest in response to AZD1775-induced RS even in cells with functional ATR.

## Deletion of DNAPK or PTEN attenuates CHK1 phosphorylation, impedes downregulation of cyclin E and overrides replication arrest in response to AZD1775 monotherapy-induced DNA damage

To investigate whether AZD1775-sensitive BLBC cells, which display loss of PTEN expression and replication-associated DNA damage, are also DNA-PK-deficient, we performed immunostaining of pT2609-DNA-PK and immunoblotting of pS2056-DNA-PK in AZD1775-sensitive, PTEN-negative HCC38 cells. As shown in *Figure 6A,B* and *Figure 6—figure supplement 1A*, AZD1775 mono- and AZD1775 + AZD6738 combination-treatment induced phosphorylation of DNA-PK in HCC38 cells,

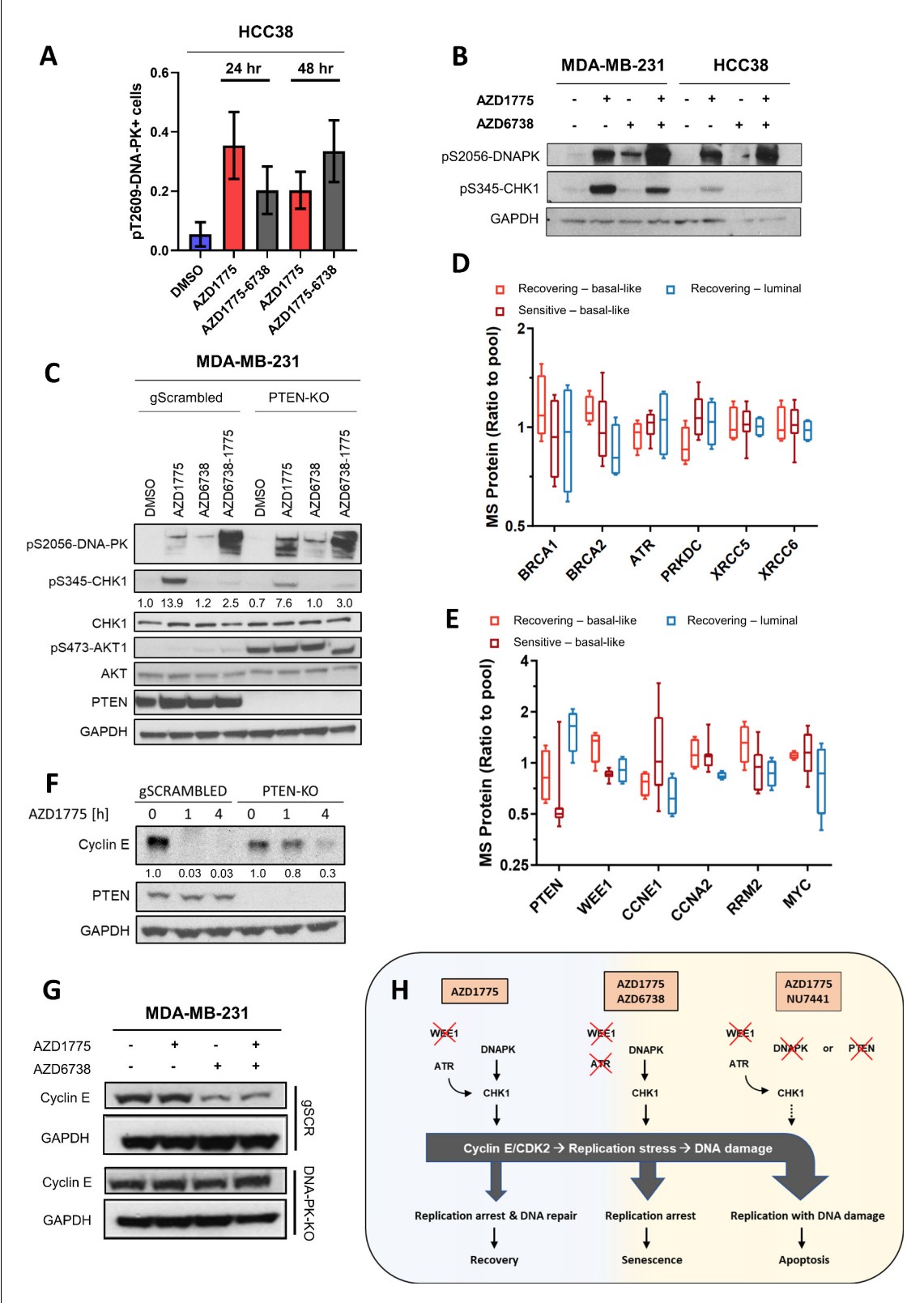

**Figure 6.** Deletion of DNAPK or PTEN attenuates CHK1 phosphorylation, impedes downregulation of cyclin E and overrides replication arrest in response to AZD1775 monotherapy-induced DNA damage. (**A**) HCC38 cells were treated with the indicated drugs for 24 or 48 hr and phosphorylation of DNA-PK (pT2609) was analysed by HCI analysis. The proportion of pT2609-DNA-PK-labelled cells represents mean of three biological replicates ± SEM. (**B**) Immunoblot analysis of CHK1 phosphorylation (pS345-CHK1) and DNAPK phosphorylation (pS2056-DNAPK) in MDA-MB-231 and

*Figure 6 continued on next page*

*Figure 6 continued*

HCC38 cells treated with the indicated drugs for 24 hr. GAPDH expression was used as loading control. (C) Phosphorylation of DNAPK (pS2056-DNAPK), CHK1 (pS345-CHK1) and AKT (pS473-AKT) in MDA-MB-231 PTEN-proficient and PTEN-deficient (PTEN-KO clone #1) treated with the indicated drugs for 24 hr. Whole cell lysates were immunoblotted with the specified antibodies. GAPDH expression was used as loading control. Expression of phosphorylated CHK1 was quantified by normalizing band intensity to CHK1 band intensity. (D) Box-plots of HR (BRCA1, BRCA2, ATR) and NHEJ (PRKDC, XRCC5, XRCC6) protein expression levels (ratio to pool) in the different groups of breast cancer cell lines according to AZD1775 response characteristics (recovering-basal, sensitive-basal and recovering-luminal). (E) Expression level (ratio to pool) box-plots of PTEN and select replication/cell cycle-associated proteins in the different groups of BC cell lines according to AZD1775 response characteristics (recovering-basal, sensitive-basal and recovering-luminal). (F) Cyclin E protein expression levels in MDA-MB-231 PTEN-proficient and PTEN-deficient (PTEN-KO clone #1) treated with AZD1775 (500 nM) for the time points indicated. Whole cell lysates were immunoblotted with the specified antibodies and quantified by ImageJ (NIH). (G) Cyclin E protein expression levels in DNA-PK-proficient and DNA-PK-deficient (DNA-PK-KO clone #1) MDA-MB-231 cells treated with the indicated drugs (500 nM AZD1775, 1 μM AZD6738) for 24 hr. Whole cell lysates were immunoblotted with the specified antibodies. (H) Schematic model of the molecular network underlying differential responses to WEE1 inhibition in BLBCs. In DNA-PK and PTEN-proficient cells, AZD1775 mono-treatment results in high levels of replication stress and activation of CHK1 to establish the replication checkpoint and to restrain CDK2 activity, resulting in replication arrest, DNA repair and recovery after drug removal (*left*). In the absence of ATR, for example due to pharmacological inhibition by AZD6738, activated DNA-PK and PTEN sustain CHK1 activation resulting in replication arrest and cellular senescence in response to AZD1775 treatment (*middle*). Conversely, in the absence or after inhibition of DNA-PK or PTEN, cells fail to elicit a checkpoint in response to AZD1775 and undergo replication catastrophe and apoptosis (*right*).

The online version of this article includes the following source data and figure supplement(s) for figure 6:

**Source data 1.** Deletion of DNAPK or PTEN attenuates CHK1 phosphorylation, impedes downregulation of cyclin E and overrides replication arrest in response to AZD1775 monotherapy-induced DNA damage.

**Figure supplement 1.** Deletion of DNAPK or PTEN attenuates CHK1 phosphorylation, impedes downregulation of cyclin E and overrides replication arrest in response to AZD1775 monotherapy-induced DNA damage.

thus ruling out that DNA-PK phosphorylation deficiency per se is the source of the replication-induced DNA damage and lack of recovery in AZD1775-sensitive cells (*Figure 1*). Although AZD1775 induced phosphorylation of CHK1 in HCC38 cells, the total level of pS345-CHK1 was lower compared to recovering MDA-MB-231 cells (*Figure 6B*), indicating that CHK1 signalling is partially compromised in cells sensitive to AZD1775 monotherapy. We next investigated whether deletion of PTEN affected AZD1775-induced DNA-PK and CHK1 phosphorylation in MDA-MB-231 cells. As shown in *Figure 6C*, phosphorylation of DNA-PK was enhanced in the PTEN-knockout cells treated with AZD1775 while S345-CHK1 phosphorylation was attenuated compared to PTEN-proficient cells. As expected, AKT was hyperphosphorylated in PTEN deleted cells (*Figure 6C*). Although these results do not establish a direct link between PTEN and DNA-PK activity, they indicate that both PTEN loss and DNA-PK loss interfere with CHK1 activation in response to AZD1775 treatment, which presumably contributes to the increased sensitivity to AZD1775 monotherapy. As low PTEN expression is one of the strongest predictors of AZD1775 response and deletion of PTEN is associated with replication-induced DNA damage (*Figure 2*), we assessed expression of proteins relevant to replication checkpoint control and AZD1775 sensitivity using our proteomic dataset. Due to the relatively small number of cell lines compared to the large number of proteins in our proteomic dataset, unbiased analysis could have a high false discovery rate and we therefore focused on specific proteins involved in DNA replication and repair. HR repair proteins ATR, BRCA1, BRCA2 or the NHEJ factors DNA-PK, XRCC5, XRCC6 did not associate with AZD1775 sensitivity, suggesting the levels of these proteins are likely not associated with AZD1775 sensitivity (*Figure 6D*). In agreement with the significant association between low PTEN protein expression and sensitivity to AZD1775, we found that the group of AZD1775-sensitive BLBC cell lines have low levels of PTEN protein compared to the group of BLBC recovering following AZD1775 monotherapy (*Figure 6E*). Notably, among the AZD1775-sensitive cell lines, only MDA-MB-157 cells had high PTEN protein levels (*Figure 6—figure supplement 1B*). Interestingly, several cell lines in the AZD1775-sensitive subgroup displayed elevated levels of cyclin E protein, including MDA-MB-157 cells harbouring CCNE1 amplification (*Figure 6E* and *Figure 6—figure supplement 1B*). In addition to low PTEN expression and/or high cyclin E expression, low WEE1 expression levels reached significance between sensitive and recovering subgroups, while neither MYC, cyclin A or RRM2 protein expression levels were associated with AZD1775 sensitivity or recovery (*Figure 6E*). Markedly, we found that AZD1775 treatment drastically reduced cyclin E protein expression in MDA-MB-231 control cells while PTEN- or DNA-PK-deleted cells exhibited only partial suppression of cyclin E in response to the AZD1775 monotherapy

(*Figure 6F,G* and *Figure 6—figure supplement 1C*). These results suggest that insufficient CHK1 activation and elevated levels of cyclin E in PTEN/DNA-PK-deleted cells could be the source of DNA damage and cell death following AZD1775 monotherapy. Supporting this possibility, knockdown of cyclin E reduced γH2AX level and increased viability in response to AZD1775 in MDA-MB-231 cells, directly linking AZD1775-induced replication stress to cyclin E expression (*Figure 6—figure supplement 1D*). Altogether, our findings support a model in which PTEN-deficient BLBC cells have a lower threshold of tolerable replication stress due to defective DNA-PK-CHK1 replication checkpoint response and elevated levels of cyclin E (*Figure 6H*).

## Discussion

WEE1 inhibitors are currently in clinical trials and identification of markers that can predict which patients could benefit from WEE1 inhibitor therapy, either as single-agent or in combination with other targeted therapies or chemotherapy, are urgently needed. Our study shows that loss of PTEN is one of the strongest markers of AZD1775 sensitivity and response in breast cancer cell lines, and that PTEN-deficient BLBC cell lines are unable to recover following removal of AZD1775. WEE1 inhibition is known to cause replication stress-driven DNA damage through over-activation of cyclin E/CDK2, which in turn triggers the ATR/CHK1 DNA damage checkpoint leading to replication arrest and DNA repair (*Sorensen and Syljuasen, 2012*). Our results show that PTEN deficiency was associated with abrogation of the replication checkpoint, resulting in continued replication despite DNA damage, leading to apoptosis (*Figure 6H*). PTEN loss deregulate PIK3/AKT/mTOR signalling, which could have an impact on replication stress for instance by increasing cyclin E levels through AKT-mediated inhibition of GSK3 (which prevent cyclin E degradation by the FBW7 ubiquitin ligase [*Arabi et al., 2012*; *Welcker and Clurman, 2008*]). However, deregulation of PTEN/PIK3/AKT signalling can also promote genomic instability through various mechanisms including inactivation of the G2 checkpoint, CHK1 phosphorylation, mis-localization of BRCA1/RPA and loading/unloading of DNA-PK onto DNA damage sites (*Song et al., 2012*; *Xu et al., 2012*). Thus, we cannot exclude the possibility that sensitivity to AZD1775 in this regard may lie beyond deregulation of AKT activity.

Interestingly, in-depth genetic analyses of breast cancer specimens pin-pointed a novel molecular breast cancer subtype characterized by a genetic footprint indicative of a defective HR-based DNA repair mechanism (*Nik-Zainal et al., 2016*). This homology repair-deficient footprint is enriched for basal-like/triple-negative characteristics, germline and somatic BRCA1 mutations, loss of p53, and notably loss of PTEN by deletion or mutation (*Nik-Zainal et al., 2016*). In addition, He et al. showed that PTEN deletion leads to dissociation of RAD51 from replication forks highlighting an important role for PTEN in regulating the replication stress cascade (*He et al., 2015*). PTEN deletion was previously shown to reduce the average inter-origin distance increasing origin density, presumably due to deregulation of dormant origin firing during RS (*He et al., 2015*). These results are in agreement with our findings that PTEN-deleted BLBCs display decreased CHK1 activation in response to AZD1775, and supporting an increased dependency on replication initiating factors such as cyclin E/CDK2. A direct link between impaired CHK1 activation, cyclin E deregulation and increased AZD1775 sensitivity in PTEN-deficient BLBC cells remains to be determined. Nonetheless, our findings are consistent with previous studies demonstrating that loss of PTEN and activation of AKT leads to reduced nuclear localization and impaired CHK1 function (*Puc et al., 2005*) and lethal interactions between inhibition of CHK1 and WEE1 in triple-negative breast cancers (TNBCs) (*Aarts et al., 2012*). Interestingly, Sen et al reported that CHK1 is activated in AZD1775-resistant cells as an escape mechanism to overcome accumulation of DNA damage and cell death, however this study did not investigate expression of PTEN/cyclin E in relation to CHK1 activation and AZD1775 sensitivity (*Sen et al., 2017*).

Prior studies have addressed the association of specific genes with acute response to AZD1775, including key regulatory proteins such as p53 (*Aarts et al., 2012*), SETD2 and RRM2 (*Pfister et al., 2015*). p53 is frequently mutated in BLBC tumours but expression of p53 was not associated with response to AZD1775 in our panel of cell lines (data not shown). As previously reported (*Pfister et al., 2015*), we found that AZD1775 treatment reduced expression of RRM2 protein, but neither RRM2 nor SETD2 expression levels significantly correlated with recovery following AZD1775 treatment although a trend towards lower RRM2 expression was found in the AZD1775 sensitive

group. In our dataset WEE1 expression levels reached significance between sensitive and recovering cell lines at 5% FDR, and interestingly, low expression of WEE1 was found in the group of AZD1775 sensitive cell lines. Whether low WEE1 expression is associated with increased cyclin E-CDK2 activity and elevated intrinsic RS, and consequently increased AZD1775 sensitivity in these breast cancer cell lines is not known. However, we did not find a significant difference of proliferation-related proteins or cell cycle populations in the different BLBC subgroups, regardless of response to AZD1775 (*Figure 6—figure supplement 1E,F and G*).

In addition to the status of PTEN as an important determinant of AZD1775 sensitivity, we also discovered a role of DNA-PK as an essential regulator of the replication checkpoint and recovery post AZD1775-treatment, linked to the activation of CHK1 independently of ATR. AZD1775 triggered ATR-CHK1 pathway activation, but also induced phosphorylation of DNA-PK independent of ATR, as DNA-PK phosphorylation was augmented when AZD1775 was combined with ATR inhibitor AZD6738. Interestingly, we found that DNA-PK knockout decreased CHK1 phosphorylation on S345 in response to WEE1 and ATR inhibition. These results underscore a more direct role for DNA-PK in regulation of the replication checkpoint in response to RS beyond its canonical function in classical NHEJ repair (*Davis et al., 2014*). In line with these results, Buisson et al. showed that activation of DNA-PK counteracts RS through a CHK1-mediated backup pathway during ATR inhibition and Lin et al. reported that DNA-PK regulates CHK1-Claspin stability for an optimal RS response (*Buisson et al., 2015*; *Lin et al., 2014*). A more direct role of DNA-PK in regulating replication origin firing has also been suggested (*Dungrawala et al., 2015*). Hence, although DNA-PK is not an essential gene, it may have indispensable checkpoint functions during high RS. Consistent with this possibility we show that DNA-PK is necessary for protection of BLBC cells from replication-associated DNA damage and maintains replication arrest following AZD1775 mono-treatment. Consequently, deletion of DNA-PK abrogates the replication block following AZD1775-induced activation of ATR-CHK1, thereby allowing replication to proceed despite DNA damage, resulting in apoptosis. Indeed, AZD1775 combined with DNA-PK inhibitors was found to be synergistic in multiple BLBC cell lines. Thus, DNA-PK activation by AZD1775 is sufficient to maintain replication arrest until the DNA damage is repaired (through ATR) and replication can restart (see model, *Figure 6H*).

Our findings that AZD1775 treatment result in rapid downregulation of cyclin E protein expression is consistent with WEE1 inhibitor-induced activation of CDK2 (loss of Y15 phosphorylation), autophosphorylation and degradation of cyclin E (*Hughes et al., 2013*). Knockdown of cyclin E desensitized BLBC cells to AZD1775, and interestingly, DNA-PK-knockout cells were unable to downregulate cyclin E in response to AZD1775 monotherapy. Correspondingly, targeted deletion of PTEN in MDA-MB-231 cells also partially prevented cyclin E downregulation. As mentioned, detailed mechanistic insights into how PTEN and DNA-PK regulate cyclin E expression in response to AZD1775 remain to be investigated. Since a common denominator for both PTEN and DNA-PK deficiency in BLBC cells was the suboptimal phosphorylation of CHK1, and since activated CHK1 is reported to inhibit cyclin/CDK activity in response to DNA damage (*Toledo et al., 2013*), one possibility is that CHK1 plays a direct function in downregulating cyclin E levels in order to maintain the replication arrest in response to AZD1775. Accordingly, Chen et al. recently reported that cyclin E overexpression is a potential biomarker predicting response to AZD1775 monotherapy in triple-negative breast cancers (TNBCs) (*Chen et al., 2018*), results consistent with our study.

In summary, our data show that both PTEN and DNA-PK status contribute to sensitivity to AZD1775 in BLBC cells. Although the relation between PTEN and DNA-PK is still unclear in this respect, both are required for activation of the CHK1 replication checkpoint, shutting down replication upon AZD1775-induced RS and shielding BLBC cells from lethal DNA damage. Future preclinical studies will determine the benefit of PTEN as a potential predictive marker of response to AZD1775 monotherapy and the role of DNA-PK in drug resistance and recurrence.

# Materials and methods

**Key resources table**

| Reagent type (species) or resource | Designation | Source or reference | Identifiers | Additional information |
|---|---|---|---|---|

*Continued on next page*

*Continued*

| Reagent type (species) or resource | Designation | Source or reference | Identifiers | Additional information |
|---|---|---|---|---|
| Genetic reagent (*Mus musculus*) | NOD/SCID | Taconic/Denmark | #NODSC | |
| Cell line (*Homo sapiens*) | MDA-MB-231 | ATCC | Cat #HTB-26 RRID:CVCL_0062 | |
| Cell line (*H. sapiens*) | MDA-MB-231 gScrambled | This Paper | | CRISPR/Cas9-expressing cell line |
| Cell line (*H. sapiens*) | MDA-MB-231 PTEN-KO | This Paper | Clone #1.4 Clone #2.3 | CRISPR/Cas9 knock-out cell lines |
| Cell line (*H. sapiens*) | MDA-MB-231 DNA-PK-KO | This Paper | Clone #1.4 Clone #2.3 | CRISPR/Cas9 knock-out cell lines |
| Cell line (*H. sapiens*) | HCC1937 -Empty vector (EV) | This Paper | | Stable pBabe-puro expressing cell line |
| Cell line (*H. sapiens*) | HCC1937-PTEN | This Paper | | Stable pBabe-puroL-PTEN expressing cell line |
| Recombinant DNA reagent | pBABE-puro (plasmid) | Addgene | RRID:Addgene_1764 | Retroviral empty backbone construct |
| Recombinant DNA reagent | pBABE-puroL PTEN (plasmid) | Addgene | RRID:Addgen 10785 | Retroviral construct expressing PTEN |
| Chemical compound, drug | AZD1775 | AstraZeneca | | |
| Chemical compound, drug | AZD6738 | AstraZeneca | | |
| Software, algorithm | GraphPad Prism 8 | GraphPad Software | Version 8.3 RRID:SCR_002798 | |
| Software, algorithm | CellProfiler | Carpenter lab, Broad Institute | Version 3.1 RRID:SCR_007358 | |
| Software, algorithm | ImageJ | Wayne Rasband, National Institutes of Health | RRID:SCR_003070 | |
| Software, algorithm | SynergyFinder | Institute for Molecular Medicine Finland | PMCID:PMC5554616 | |
| Software, algorithm | Depmap portal | Broad Institute | RRID:SCR_017655 | |

## Cell culture

Breast cancer cell lines included in this study were the luminal cell lines HCC1419, MCF7, CAMA, HCC1428, T47D, and the basal-like cell lines HCC38, HCC70, HCC1569, HCC1937, MDA-MB-157, HCC1143, BT20, MDA-MB-231, MDA-MB-468, BT549, SUM159PT, Cal51, HS578T and SUM149PT. All cell lines were purchased from the American Type Culture Collection (ATCC) except SUM149PT, which was obtained from Biovit (https://bioivt.com/sum-breast-cancer-cell-lines). Cell lines were maintained in RPMI1640 (Gibco, Thermo Fisher) supplemented with 10% FBS (Gibco, Thermo Fisher) and 2 mM L-glutamine (Gibco, Thermo Fisher), 10 mM HEPES (Gibco, Thermo Fisher), and 1 mM Sodium Pyruvate (Gibco, Thermo Fisher). All cell lines were routinely tested for mycoplasma infection and authenticated using STR profiling.

## Antibodies

All the antibodies are listed in *Supplementary file 4*.

## Plasmids, lentiviruses and generation of CRISPR/Cas9 knockout and PTEN reconstituted stable cell lines

To generate stable CRISPR/Cas9 knockout cell lines, sgRNAs targeting selected genes were first synthesized and cloned into the pLenti Cas9-puromycin v.2.0. The Broad Institute sgRNA algorithm was used to design a minimum two sgRNA per targeted genes. Lentiviruses were generated in HEK293FT cells by co-transfecting the specified sgRNA containing-pLenti Cas9 v.2.0 together with

the lentivector packaging plasmids, psPAX2 and pMD2.G. Supernatants containing viral particles (48–96 hr post transfection) were pooled and used to infect target cells in the presence of freshly added hexadimethrine bromide (10 mg/mL). Seventy-two hours post-infection, the transduced cells were subsequently selected and maintained as pools or clones in the presence of puromycin (0.5–1.0 µg/mL). Knockdown was assessed by immunoblotting. To verify genomic editing in DNA-PK isogenic knockout cell lines the targeted region was PCR-amplified and cloned into pTOPO2.1, followed by Sanger-sequencing of at least six clones per cell line. If all sequencing reactions revealed the same edited allele in a cell line, we additionally confirmed the presence of only one allele by using Alt-R genome editing detection kit (IDT) according to manufacturer's instructions. The lentiviral plasmid for expression of PTEN was a gift from William Sellers (Addgene plasmid #10785). Lentiviruses were generated and cells transduced as described above for knockout cell lines. Expression was verified by immunoblot analysis.

## High-Content siRNA screening

High content siRNA screens were performed in 384-well format. Briefly, the DNA damage siRNA library consisting of three individual siRNAs for 300 DNA repair genes (Dharmacon) was preprinted into 384 wells in final concentration of 17 nM and complexed with 0.06 µL of siLentFect (Bio-Rad) lipid transfection agent. 1250 MDA-MB-231 (ATCC) cells were applied onto each 384 well as a suspension and allowed to transfect for 24 hr. Replicate one screen plates were treated with 500 nM AZD1775 and replicate 2 plates with 0,01% DMSO. After 48 h cells were fixed with 2% paraformaldehyde and permeabilized with 0.3% Triton-X 100% and 10% horse serum in PBS and blocked with 10% horse serum. Cells were assayed with antibody for gamma-H2AX (Abcam, Cambridge, UK) and labelled with goat-anti-mouse Alexa647 (Molecular Probes, Carlsbad, CA, USA) secondary antibody and DAPI for DNA. Immunostained cells were analysed by microscopic imaging, using an Olympus scanR high content imager (Olympus, Hamburg, Germany) equipped with a Hamamatsu ORCA-ER CCD digital camera (Hamamatsu Photonics K.K., Hamamatsu, Japan). Cells were analysed using the scanR image analysis software. Integrated nuclear DNA staining was used for imaging cytometry analysis of cell cycle distribution and gamma-H2AX foci counts per nuclei were quantified using a watershed object identification algorithm. For analysis of significance, a Z-score was calculated for cell counts as measure of viability and gamma-H2AX foci counts for induction of DNA damage using global plate mean values and standard deviation for all samples in the plate, including negative controls and excluding positive controls. From this analysis, siRNAs with Z-scores ± 2 standard deviations were considered significant. siRNA DNA repair pathway-based pattern identification analysis was performed using gene ontology annotations from AmiGO two gene ontology database (http://amigo.geneontology.org/amigo/landing). RNAi data have been deposited at Mendeley (https://data.mendeley.com/) under DOI 10.17632/rmjnmwzmf6.1.

## High-Content imaging (HCI) drug screening

Microscopic imaging-based drug screening with breast cancer cell lines was performed using an Olympus scanR integrated high content imager and image analysis suite (Olympus-SIS). AZD1775 and AZD6738 was tested with six 3-fold dilutions starting from 10 µM and 20 µM as the highest concentration, respectively. Each 384-well sample well was imaged with a 10 × objective using specific filter sets for DAPI (Semrock, Inc). The effect of the drugs on cell numbers as indicator of cell viability was assessed by comparing cell counts measured in DMSO (negative controls). The DNA counterstaining of the cells was performed according to the following protocol. First the culture medium was aspirated carefully from each well and the cells were fixed with 2% paraformaldehyde (Sigma-Aldrich) in PBS for 15 min at room temperature. Cells were then washed once for 5 min with PBS. Cell permeabilized was done with 0.3% Triton-X100 in 20 µL of PBS for 15 min at room temperature, followed by PBS wash. DAPI (4',6-diamidino-2-phenylindole) DNA counterstaining was performed for 1 hr at room temperature, followed by washing with PBS.

## High-content immunofluorescence imaging and drug response assessment

AZD1775 (MK-1775/Adavosertib) and AZD6738 (Ceralasertib) were acquired from Astra Zeneca, whilst DNA-PK inhibitor NU-7441 and ATM inhibitor KU60019 were purchased from Selleckchem.

Initial dose response was assessed by image cytometry of 384-wells plates containing cells treated with increasing doses of drugs for 72 hr. At the treatment endpoint, the cells were fixed with 4% paraformaldehyde, permeabilized with 0.4% Triton X-100, stained with DAPI (1 µg/ml) and labelled with the indicated antibodies in TBS- 0.1% Tween 20 overnight. Non-conjugated primary antibodies were counterstained with the appropriate fluorochrome-conjugated secondary antibodies (Alexa Fluor) for 2 hr prior to further wash in TBS and image acquisition using Olympus ScanR at 10x magnification. Minimum four tiles were captured per well for quantitative analysis using Olympus ScanR Analysis Software: the DAPI-positive region of each cell was used as a boundary to quantitate nuclear signal, and a 10 pixel annulus around the nucleus was used to quantitate cytoplasmic signal, omitting nuclear signal. To assess phosphorylation of γH2AX (pS139), DNA-PK (pS2056 and pT2609) and CHK1 (pS345) and expression of DNA-PK, CHK1 and RAD51 by high-content imaging, cells were plated in clear-bottom 96-well plates (Corning) treated as indicated depending on the experimental setup and subsequently stained as described above. After final washes cells were overlaid with PBS, imaged using an ImageXpress µ automated microscope (Molecular devices) and images analysed using CellProfiler software (*Carpenter et al., 2006*).

Validation of dose responses were performed using either crystal violet or alamarBlue (Thermo-Fisher) on 96-well plates. Cells were seeded 24 hr prior to treatment with increasing doses of single drug or combination for 72 hr. At treatment endpoint, cells were fixed with 100% methanol and stained with 0.05% crystal violet for spectrophotometric absorbance-based at 592 nm quantification of cell number. When assessing cell proliferation by alamarBlue assay, the reagent was added directly to the cell medium and fluorescence measured according to manufacturer's instructions at time point 0 and again after 3 hr incubation at 37°C. Where indicated, crystal violet staining assays and quantifications were performed on 24-well and 6-well plates in triplicates.

## Western blot analysis

For biochemical analyses of total protein expression and phosphorylation, cells were lysed in NP-40 lysis buffer (50 mM Tris-HCl pH 8.0, 150 mM NaCl, 1% NP-40) supplemented with both protease (Complete mini, Roche) and phosphatase (PhosSTOP, Roche) inhibitors. Proteins were resolved in 4–12% Bolt Bis-Tris Plus gel (Invitrogen) under reducing-denaturing condition, transferred to PVDF membranes (Bio-Rad), blocked by 5% nonfat milk or BSA in TBS-T, and immunoblotted with the indicated antibodies. Detection was performed by Western Lightning Plus-ECL Substrate Kit according to the manufacturer's instructions (PerkinElmer). Where required to clarify relative differences, the resulting signals were quantified using ImageJ software.

## Cell cycle progression and cell death assessment by flow cytometry

To measure the effects on cell cycle progression and DNA replication, cells were treated with the indicated compounds for the times specified. 10 µM EdU for 1 hr at 37 °C was used to label actively replicating cells. Depending of the purpose of experiment, EdU-labelling of the cells was performed either prior to treatment or 1 hr prior to fixation for further processing according to the kit protocol (Molecular ProbesClick-IT Plus Edu Alexa Fluor 488 or 647 Flow Cytometry Assay Kit, Invitrogen). For labelling of γH2AX, fixed and permeabilized cells were stained with γH2AX pS139 (1:500, CST), and 50 µg / mL Propidium Iodide in 1% BSA/TBS – 0.5% Tween 20 containing 10 µg/mL RNAseA. The click-iT reaction was carried out according to the manufacturer's protocol prior to FACS analysis. For analysis of apoptosis, treated cells were trypsinized and live-stained with Annexin V-FITC and Propidium Iodide according manufacturer's protocol (ThermoFisher). Flow cytometry data were acquired using a FACSCanto II flow cytometer (BD) and analysed using FACSDiva software (BD).

## SA-βgal staining assay

Staining to detect senescence-associated beta-galactosidase (SA-βgal) activity was carried out as described previously (*Debacq-Chainiaux et al., 2009*). Briefly, cells were plated on 12-well plates, treated as indicated, subsequently fixed in 4% paraformaldehyde for 10 min, then incubated at 37°C for 12 to 24 hr in staining solution containing 1 mg/ml X-gal (Sigma-Aldrich). Images were taken on a Leica DMI600 microscope and analysed using CellProfiler software (*Carpenter et al., 2006*).

## Orthotopic xenografts

All animal protocols in this study were approved by the ethical committee for animal experiments of northern Stockholm (N2451/15). Mice were maintained under pathogen-free conditions according to guidelines of the animal facility at MTC, Karolinska Institutet. 6–8 weeks old NOD/SCID female mice (Taconic/Denmark) were injected with $1 \times 10^6$ MDA-MB-231 cells orthotopically in the mammary gland #4. The injected volume was 0.1 ml and the mice were sedated with isoflurane during the injection. The cells had been grown in culture in RPMI 10% FBS and harvested in the growth phase before reaching confluency on the day of injection and they were washed twice with PBS and counted under the inverted microscope using a hematocytometer. When mice started developing tumours (avg initial tumour volume = 67,92 + 10,66 mm3 (avg + sem) mice started receiving one of the following treatments: vehicle (n = 4), AZD6738 25 mg/kg (n = 4), AZD1775 25 mg/kg (n = 4) or AZD6738 25 mg/kg + AZD1775 25 mg/kg ('combo') (n = 6). The volume administered was 0,1 ml and the vehicle used was: DMSO 4%/PEG 30%/Tween80 5% in water. Treatment was administered orally by gavage, 5 days a week, for 12 days (5 days on, 2 days off, 5 days on). Mice were inspected daily and tumours were measured with a caliper. Tumour volume was calculated as: $V = D \times d^2 \times \pi$. Mice started receiving one of the following treatments: vehicle (n = 4), AZD6738 25 mg/kg (n = 4), AZD1775 25 mg/kg (n = 4) or AZD6738 25 mg/kg + AZD1775s a control for any possible toxicity and the body weight was recorded. When tumours reached 1000 mm3 mice were terminated and the tumours collected and divided in three fragments: one was frozen in OCT (Cryomount, Histolab); the other was fixed in buffered 4% formaldehyde solution (Histolab) and paraffin-embedded and the last one was snap-frozen. Mice received the last dose of treatment 3 hr before termination. An i.p. injection of BrdU (0,1 ml of a 10 mg/ml solution, Sigma) was done 1 hr before termination, to be used as a marker of cell proliferation. Apart from the tumours, lungs and skin were also collected for evaluation of metastases and the as a control of proliferative epithelia respectively.

## Immunohistochemistry and immunofluorescence staining of tumour tissues

The detection of BrdU, Ki67 and phospho-DNA-PK was done on formalin-fixed, paraffin-embedded tumour tissue from the xenograft experiment, using sections of 4 microns thickness. Citrate buffer pH6 was used as antigen-retrieval method for all stainings. For the BrdU staining, an additional step with DNAse I incubation was used. Primary antibodies used were: rat anti-BrdU (Abcam, ab6326), rabbit anti-Ki67 (Abcam, ab16667) or rabbit anti-DNA-PK (phospho-S2056) (Abcam, ab18192). Detection: BrdU and Ki67 were detected by IHC using an HRP-conjugated secondary antibody (Vector; 1:500) and DAB (Vector). Hematoxylin (Histolab) was used as counterstaining. The whole slides were scanned using a slide scanner and the images were opened in the image software Case Viewer and zoomed 20 times (three tumours per group were analysed). The detection of phopho-DNA-PK was done by immunofluorescence using an Alexa Fluor 555 goat anti-rabbit (Invitrogen, A21428) as secondary antibody and DAPI as nuclear counterstaining. Fluorescent images were taken in a Zeiss LSM880 confocal microscope at a magnification of 63X using immersion oil. The images were taken as stacks. Three microscopic fields per tumour were imaged using the 555 nm ('red') and 405 nm ('blue').

## Proteome profiling of breast cancer cell lines

### Sample preparation

Biological triplicates of breast cancer cell lines (SKBR3, MCF7, LCC2, HCC70, HCC1954, HCC1937, HCC1569, HCC1187, BT549, T47D, MDAMB157, CAL51, SUM149PT, HCC1143, BT20, HCC38, HCC1419) were harvested and cell pellets lysed in 4% SDS, 25 mM HEPES pH 7.6 and 1 mM DTT. Total protein amount was estimated (Bio-Rad DC). Protein digestion (LysC and trypsin, sequencing grade modified, Pierce) was performed using a modified SP3-protocol (*Hughes et al., 2014*). In brief, each sample was reduced with 1 mM DTT and alkylated with 40 mM Chloroacetamide. Sera-Mag SP3 bead mix (20 µl) was transferred into the protein sample together with 100% Acetonitrile to a final concentration of 70%. The mix was incubated under rotation at room temperature for 18 min. The mix was placed on the magnetic rack and the supernatant was discarded, followed by two washes with 70% ethanol and one with 100% acetonitrile. The beads-protein mixture was reconstituted in 100 µl LysC buffer (0.5 M Urea, 50 mM HEPES pH: 7.6 and 1:50 enzyme (LysC) to protein

ratio) and incubated O/N. Finally, trypsin was added in 1:50 enzyme to protein ratio in 100 µl 50 mM HEPES pH 7.6 and incubated O/N. The peptides were eluted from the mixture after placing the mixture on a magnetic rack, followed by peptide concentration measurement (Bio-Rad DC Assay). Before labelling, samples were pH adjusted using TEAB pH 8.5 (50 mM final conc.). 100 µg of each sample were labelled with an isobaric TMT-tag (Thermo Scientific). Labelling efficiency was determined by LC-MS/MS before pooling of samples. Sample clean-up was performed by solid phase extraction (SPE strata-X-C, Phenomenex). Purified samples were dried in a SpeedVac.

## High resolution isoelectric focusing (HiRIEF)

After pooling and sample clean-up by solid phase extraction (SPE strata-X-C, Phenomenex), the sample pool was subjected to peptide IEF-IPG (isoelectric focusing by immobilized pH gradient) in pI range 3–10 (350 µg). Freeze dried peptide samples were dissolved in 250 µL rehydration solution containing 8 M urea and 1% IPG pharmalyte pH 3–10 and allowed to adsorb to the 24 cm linear gradient IPG strip by swelling overnight. Peptides were focused on the IPG strip as described in *Branca et al., 2014*. After focusing, the peptides were passively eluted into 72 contiguous fractions with MilliQ water using an in-house constructed IPG extractor robotics (GE Healthcare Bio- Sciences AB, prototype instrument) into a 96-well plate (V-bottom, Corning product #3894), which were then dried in a SpeedVac. The resulting fractions were freeze dried and kept at −20˚C.

## LC-MS/MS analysis

Online LC-MS was performed using a hybrid Q-Exactive - HF mass spectrometer (Thermo Scientific). For each LC-MS/MS run, the auto sampler (Dionex UltiMate 3000 RSLCnano System) dispensed 20 µl of solvent A to the well in the 96 V plate, mixed, and proceeded to inject 10 µl. FTMS master scans with 70,000 resolution (and mass range 300–1700 m/z) were followed by data-dependent MS/ MS (35 000 resolution) on the top five ions using higher energy collision dissociation (HCD) at 30– 40% normalized collision energy. Precursors were isolated with a 2 m/z window. Automatic gain control (AGC) targets were 1e6 for MS1 and 1e5 for MS2. Maximum injection times were 100 ms for MS1 and 150–200 ms for MS2. The entire duty cycle lasted ~2.5 s. Dynamic exclusion was used with 60 s duration. Precursors with unassigned charge state or charge state one were excluded. An underfill ratio of 1% was used. Spectra data were converted to mzML files using ProteoWizard release: 3.0.10827 (2017-5-11) and searched with MS-GF+ (2016.10.26) (*Kim and Pevzner, 2014*) and Percolator (*Käll et al., 2007*). Precursor mass tolerance used was 10 ppm, fragment mass tolerance 0.11 Da, fixed modifications were TMT-10plex on lysines and peptide N-termini, and carbamidomethylation on cysteine residues, oxidation on methionine was used as a variable modification. The protein database used for search was Ensembl version 75 human protein databases (104,763 protein entries) allowing for one tryptic miss-cleavage. A pool of all samples was used in one TMT tag as linker (denominator) between TMT sets. Labelling scheme can be found together with raw and search result data with dataset identifier PXD013276. On average 14 unique peptides per protein and on average 45 PSMs per protein were used for quantification, resulting in a correlation between replicates (same gene across all cell lines) of above 0.9 for 75% of the data (*Figure 2—figure supplement 1B*). The intensity of internal reference (TMT tag 131) was used as denominator to calculate peptide ratios, the median of peptide ratios was taken as protein ratios. PSMs and proteins were filtered at 1% PSMs and protein level FDR. The mass spectrometry proteomics data have been deposited to the ProteomeXchange Consortium via the JPOST partner repository with the dataset identifier PXD013276. All data are preserved for transparency as source data files.

## Transcriptomic analysis

RNA library preparation was done using Illumina TruSeq Stranded total RNA and Illumina RiboZero. Paired-end RNA sequencing was performed using HiSeq2500 on 18 breast cancer cell lines generating on average 85 million uniquely mapped reads per sample at Genomics facility, Science for Life Laboratory, Karolinska Institutet, Sweden. Transcriptome data have been uploaded to GEO (GSE152102) and processed data are provided as *Supplementary file 3* and source data file for transparency. In order to compare the quantitative proteomics and transcriptomics data quantifications, both data types were collapsed to gene symbol centric measurements. For transcriptomic analysis, the breast cancer cell line assays were performed in biological triplicates.

## Statistical analysis

Statistical tests were performed using the statistical package GraphPad Prism 8 (GraphPad Software) using either Student's t-test (p-value<0.05 was considered to be statistically significant), Log-rank (Mantel-Cox), one-way analysis of variance (ANOVA) or two-way analysis of variance (ANOVA) with Bonferroni's *post hoc* test where appropriate. For correlation analysis both Spearman's correlation as well as Pearson's correlation coefficients along with two-tailed p-values were calculated. Error bars represent standard deviation (SD) or standard error of the mean (SEM) as indicated in the figure legends. All experiments were repeated at least three times if not otherwise stated.

## Acknowledgements

The AZD1775, AZD6738, AZD5363 inhibitors was provided by AstraZeneca. We thank Alexandros Kourtesakis and Aljona Maljukova for technical assistance, Kashyap Dave for helpful discussions, and Arne Lindqvist for critical reading of the manuscript. This research was funded by The Swedish Cancer Society, The Swedish Research Council, Karolinska Institute, Radiumhemmets Research Foundation and in part by the AstraZeneca-SLL-KI Open Innovation grant #18122013.

## Additional information

### Funding

| Funder | Grant reference number | Author |
| --- | --- | --- |
| Cancerfonden | CAN2017/326 | Andrä Brunner<br>Olle Sangfelt |
| Vetenskapsrådet | 2014-3405 | Olle Sangfelt |
| Karolinska Institutet | | Olle Sangfelt |
| Radiumhemmets Forsknings-fonder | 171183 | Aldwin Suryo Rahmanto<br>Olle Sangfelt |
| AstraZeneca | AstraZeneca-SLL-KI Open Innovation grant #18122013 | Andrä Brunner<br>Aldwin Suryo Rahmanto<br>Henrik Johansson<br>Marcela Franco<br>Johanna Viiliäinen<br>Oliver Frings<br>Erik Fredlund<br>Charles Spruck<br>Janne Lehtiö<br>Juha K Rantala<br>Lars-Gunnar Larsson<br>Olle Sangfelt<br>Mohiuddin Gazi |
| Barncancerfonden | TJ2018 | Johanna Viiliäinen<br>Olle Sangfelt |
| Cancerfonden | CAN2017/781, 19 0561 Pj | Lars-Gunnar Larsson |
| Barncancerfonden | PR2017-0161 | Lars-Gunnar Larsson |
| Karolinska Institutet | | Lars-Gunnar Larsson |
| Radiumhemmets Forsknings-fonder | 191182 | Lars-Gunnar Larsson |

The funders had no role in study design, data collection and interpretation, or the decision to submit the work for publication.

### Author contributions

Andrä Brunner, Data curation, Formal analysis, Validation, Investigation, Visualization, Methodology, Writing - original draft, Writing - review and editing; Aldwin Suryo Rahmanto, Formal analysis, Validation, Investigation, Methodology; Henrik Johansson, Resources, Data curation, Formal analysis, Visualization, Methodology; Marcela Franco, Resources, Investigation, Visualization; Johanna

Viiliäinen, Validation, Investigation, Visualization, Writing - review and editing; Mohiuddin Gazi, Investigation; Oliver Frings, Resources, Data curation, Software, Formal analysis, Visualization; Erik Fredlund, Resources, Data curation, Software, Formal analysis, Funding acquisition, Visualization, Methodology; Charles Spruck, Conceptualization, Funding acquisition, Writing - review and editing; Janne Lehtiö, Conceptualization, Supervision, Funding acquisition, Methodology; Juha K Rantala, Conceptualization, Resources, Data curation, Software, Formal analysis, Funding acquisition, Investigation, Visualization, Methodology; Lars-Gunnar Larsson, Conceptualization, Resources, Funding acquisition, Writing - original draft, Writing - review and editing; Olle Sangfelt, Conceptualization, Formal analysis, Supervision, Funding acquisition, Visualization, Writing - original draft, Project administration, Writing - review and editing

### Author ORCIDs

Andrä Brunner (iD) https://orcid.org/0000-0003-4316-5713
Aldwin Suryo Rahmanto (iD) http://orcid.org/0000-0002-4593-286X
Marcela Franco (iD) http://orcid.org/0000-0001-8444-1708
Johanna Viiliäinen (iD) https://orcid.org/0000-0002-5527-4266
Janne Lehtiö (iD) http://orcid.org/0000-0002-8100-9562
Lars-Gunnar Larsson (iD) https://orcid.org/0000-0001-6914-3147
Olle Sangfelt (iD) https://orcid.org/0000-0002-0316-0195

### Ethics

Animal experimentation: All animal protocols in this study were approved by the ethical committee for animal experiments of northern Stockholm (N2451/15).

### Decision letter and Author response

Decision letter https://doi.org/10.7554/eLife.57894.sa1
Author response https://doi.org/10.7554/eLife.57894.sa2

## Additional files

### Supplementary files

- Supplementary file 1. AZD1775 response characteristics in human breast cancer cell lines.
- Supplementary file 2. Transcriptomics data for human breast cancer cell lines.
- Supplementary file 3. Proteomics data for human breast cancer cell lines.
- Supplementary file 4. List of all antibodies.
- Transparent reporting form

### Data availability

All data generated or analyzed during this study are included in the manuscript and supporting files. MS proteomics data were uploaded to ProteomeXchange under the identifier PXD013276, RNA-seq data are deposited on NCBI GEO under the identifier GSE152102 and siRNA drug response screen data are available on Mendeley, DOI https://doi.org/10.17632/rmjnmwzmf6.1.

The following datasets were generated:

| Author(s) | Year | Dataset title | Dataset URL | Database and Identifier |
|---|---|---|---|---|
| Brunner A, Rahmanto AS, Johansson H, Franco M, Viiliäinen J, Mohiuddin G, Frings O, Fredlund E, Spruck C, Lehtiö J, Rantala JK, Larsson LG, Sangfelt O | 2020 | Proteome characterization of Breast Cancer Cell lines | http://proteomecentral.proteomexchange.org/cgi/GetDataset?ID=PXD013276 | ProteomeXchange, PXD013276 |

| Brunner A, Rahmanto AS, Johansson H, Franco M, Viiliäinen J, Mohiuddin G, Frings O, Fredlund E, Spruck C, Lehtiö J, Rantala JK, Larsson LG, Sangfelt O | 2020 | Transcriptome analysis of Breast Cancer Cell lines | https://www.ncbi.nlm.nih.gov/geo/query/acc.cgi?acc=GSE152102 | NCBI Gene Expression Omnibus, GSE152102 |
| Brunner A, Rahmanto AS, Johansson H, Franco M, Viiliäinen J, Mohiuddin G, Frings O, Fredlund E, Spruck C, Lehtiö J, Rantala JK, Larsson LG, Sangfelt O | 2020 | High-content imaging RNAi drug response screen | https://doi.org/10.17632/rmjnmwzmf6.1 | Mendeley, 10.17632/rmjnmwzmf6.1 |

The following previously published datasets were used:

| Author(s) | Year | Dataset title | Dataset URL | Database and Identifier |
| --- | --- | --- | --- | --- |
| Nusinow DP, Szpyt J, Ghandi M, Rose CM, McDonald ER, Kalocsay M, Jané-Valbuena J, Gelfand E, Schweppe DK, Jedrychowski M, Golji J, Porter DA, Rejtar T, Wang yK, Kryukov GV, Stegmeier F, Erickson BK, Garraway LA, Sellers WR, Gygi SP | 2020 | Normalized protein expression data for all cell lines | https://gygi.med.harvard.edu/publications/ccle | Depmap project portal, ccle |
| McFarland JM, Ho ZV, Kugener G, Dempster JM, Montgomery PG, Bryan JG, Krill-Burger JM, Green TM, Vazquez F, Boehm JS, Golub TR, Hahn WC, Root DE, Tsherniak A | 2018 | DEMETER2 data v.6 | https://ndownloader.figshare.com/files/13515380 | Depmap project portal, 13515380 |

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
