## [Decision Letter]

[Editors' note: this paper was reviewed by Review Commons.]

**Acceptance summary:**

PTEN is frequently mutated in human cancers, and most often the loss of PTEN confers drug resistance in those tumors. In this manuscript the authors find that PTEN loss sensitizes basal-like breast cancer lines to WEE1 inhibitors. Moreover, they find that WEE1 inhibitors synergize with DNA-PK inhibitors to kill this tumor type. The work has important clinical ramifications for the therapy of this and other cancers.

**Decision letter after peer review:**

Thank you for submitting your article "PTEN and DNA-PK determine sensitivity and recovery in response to WEE1 inhibition in breast cancer" for consideration by *eLife*. Your article has been reviewed by three peer reviewers, and the evaluation has been overseen by Maureen Murphy as the Senior and Reviewing Editor.

The reviewers have discussed the reviews with one another and the Reviewing Editor has drafted this decision to help you prepare a revised submission.

As the editors have judged that your manuscript is of interest, but as described below that additional experiments are required before it is published, we would like to draw your attention to changes in our revision policy that we have made in response to COVID-19 (https://elifesciences.org/articles/57162). First, because many researchers have temporarily lost access to the labs, we will give authors as much time as they need to submit revised manuscripts. We are also offering, if you choose, to post the manuscript to bioRxiv (if it is not already there) along with this decision letter and a formal designation that the manuscript is 'in revision at *eLife*'. Please let us know if you would like to pursue this option. (If your work is more suitable for medRxiv, you will need to post the preprint yourself, as the mechanisms for us to do so are still in development.)

In this manuscript the authors identify two proteins whose presence dictates resistance to WEE1 inhibitors: PTEN and DNA-PK. The former could serve as a biomarker, and the latter could serve as an actionable target for breast cancer. The authors perform an analysis of AZD1775 sensitivity in relation to the proteome and transcriptome in a panel of breast cancer cell lines. They identify a novel synthetic lethal interaction between DNA-PK and the WEE1 inhibitor AZD1775 in basal-like breast cancer cells. The authors create two CRISPR knockout clones of MDA-MB-231 cells and show that loss of PTEN confers sensitivity to WEE1 inhibition by AZD1775. In an odd turn, the authors then turn to combinations of AZD1775 with an ATR inhibitor, and they show the combination is more effective in 231 xenografts. The authors then perform a high-content image-based RNA interference (RNAi) screen of 300 DDR genes in MDA-MB-231 cells, to search for other targets. They identify DNA-PK as a potential target, and they claim to show that NU7441 and AZD1775 are synergistic against basal like breast cancer with high PTEN, but these data do not seem to appear in any of the (several) submitted versions of the manuscript. Overall the manuscript is interesting and the data in it extend upon the findings in the literature of the impact of ATR plus WEE1 inhibitors.

Overall, if the authors are able to complete the following, this work could be suitable for *eLife*:

1) The authors should show a table of combination indices which demonstrates synergy between DNA-PK inhibitor NU7441 and AZD1775. Ideally this should also be accompanied by a xenograft analysis of this combination, similar to what the authors have done for ATR/WEE1 combination in Figure 3, but in these days of quarantine it will be acceptable to show rigorous evidence for synergy of NU7441 and AZD1775 in multiple BLBC lines.

2) Much of the work has been done on a single WEE1 inhibitor. In the interest of rigor, if the authors can show that silencing WEE1 or use of an independent WEE1 inhibitor also shows reliance on PTEN loss for sensitivity, this would improve the broadness of the conclusions.

3) The model shown in Supplementary figure 7 should be condensed and shown as a main figure in the manuscript.

4) Finally, the authors should go through the manuscript with a fine-tooth comb and make sure that each figure is referenced correctly, and in order, in the manuscript. Unfortunately, the copy of the revised manuscript did not have figures attached, so it was impossible to do this myself, and several of the figures seemed inaccurately referenced in the text of the manuscript.

---

## [Author Response]

Overall, if the authors are able to complete the following, this work could be suitable for eLife:1) The authors should show a table of combination indices which demonstrates synergy between DNA-PK inhibitor NU7441 and AZD1775. Ideally this should also be accompanied by a xenograft analysis of this combination, similar to what the authors have done for ATR/WEE1 combination in Figure 3, but in these days of quarantine it will be acceptable to show rigorous evidence for synergy of NU7441 and AZD1775 in multiple BLBC lines.

We have now performed additional drug synergy analysis of DNA-PK and WEE1 inhibitors. The results demonstrate potent synergistic growth inhibition activity for the AZD1775-NU7441 combination in multiple BLBC cell lines (new Figure 4C and new Figure 4—figure supplement 1B).

2) Much of the work has been done on a single WEE1 inhibitor. In the interest of rigor, if the authors can show that silencing WEE1 or use of an independent WEE1 inhibitor also shows reliance on PTEN loss for sensitivity, this would improve the broadness of the conclusions.

To explore this important issue in more detail, we used the DepMap portal (https://depmap.org/portal) to estimate WEE1 gene dependency distribution in relation to PTEN protein expression in a large number of breast cancer cell lines. The results from this analysis both validate and expand our findings that low PTEN protein level is a potential vulnerability in WEE1-targeted breast cancer cells. Thus, a greater dependency on WEE1 was found in breast cancer cells with low PTEN levels as shown by a significant correlation between PTEN protein expression and WEE1 gene effect in two different proteome datasets (this study and Nusinow et al., see new Figure 2C and new Figure 2—figure supplement 2C). As suggested, we confirmed that an independent WEE1 inhibitor (PD0166285) reduced cell viability in a PTEN-dependent manner, similar to AZD1775 (see new Figure 2—figure supplement 2E). Finally, we demonstrate reliance of PTEN for sensitivity to WEE1 depletion in both PTEN-KO (MDA-MB-231 cells) and PTEN-restored cells (HCC1937) (new Figure 2—figure supplement 3).

3) The model shown in Supplementary figure 7 should be condensed and shown as a main figure in the manuscript.

We have condensed the model which is now incorporated as a main Figure 6H.

4) Finally, the authors should go through the manuscript with a fine-tooth comb and make sure that each figure is referenced correctly, and in order, in the manuscript. Unfortunately, the copy of the revised manuscript did not have figures attached, so it was impossible to do this myself, and several of the figures seemed inaccurately referenced in the text of the manuscript.

We are sorry for this. As pointed out, we have carefully examined the revised manuscript for any inaccurately referenced figures or other misprints. A new revised version with figures attached has been submitted.

Below follows the previous letter of response to the reviewer’s comments with minor modifications.

General response to reviewer #1.

We are glad the reviewer finds our manuscript potentially interesting and significant, but troubled by the implications that the data is difficult to interpret. However, we do acknowledge and apologize that the quality of some of the immunofluorescence images provided in the first version of the manuscript were of low resolution/magnification, this has now been amended. We also paid attention to the reviewer’s concern regarding incomplete material, methods and figure legends, and have made appropriate corrections throughout the manuscript. We thank the reviewer for the thorough revision of our paper and have addressed all the different points raised, either by additional experiments or clarification/discussion to the best of our ability as specified in detail below.Reviewer #1 (Evidence, reproducibility and clarity):Summary:By doing global proteome and transciptome analysis on a panel of breast cancer cell lines, the authors identified two response group to AZD1775 in basal-like breast cancer (BLBC) cells separated by their PTEN status. The authors defined BLBC cells as recurrent if the cell lines are able to recover after removal of AZD1775. The authors found that PTEN high BLBCs were able to recover from AZD1775 treatment. Since AZD1775 induced DNA-PKc phosphorylation, the authors proposed that it protected cells from replication associated DNA damage and promoted cell recovery. Using CRISPR/Cas9 to knockout DNA-PKc or PTEN, the authors showed that it sensitized the recurrent BLBCs to AZD1775. The authors concluded that PTEN would be a promising biomarker for stratifying breast cancer patients for WEE1 cancer therapy. The manuscript contains an impressive amount of data, however,the quality of the figures, theincomplete Material and methods, theinadequate figure legendsetc. make it impossible to interpret. I have included a list of concerns which is by no means complete. The manuscript requires extensive re-organization.Comments:All the immunofluorescence images provided in this manuscript are low resolution, low magnification, and blurry(Figure 3E, 5C, and Figure 3—figure supplement 2C, Figure 4—figure supplement 1B-C).As such, it is extremelydifficult to interpret either cell morphology or sub-cellular localizationof proteins. In some cases,DAPI is too faint to see(Figure 5C). Furthermore,scale bars are missingin several images (Figure 3—figure supplement 1B, Figure 4—figure supplement 1D-E) and in other cases scale bars are present but the reference size is not providedin the legend (Figure 3E, 5C and Figure 3—figure supplement 1D, Figure 3—figure supplement 2C, Figure 4—figure supplement 1B).Higher resolution and higher magnification images are required.If any immunofluorescentimages are stitched together(e.g. Figure 3—figure supplement 2C), it should bestated in the legend.We apologize that the immunofluorescence images were of poor quality in the original submission and now provide higher resolution and magnifications, including scale bars with reference size. All merged images are clearly separated by lines which is also stated in corresponding legends.Control (total DNAPK levels) is missing in the western blots (Figure 6B and C) and immunofluorescence images (Figure 5C).Similar comment for pS345-CHK1 (Figure 6C).

We have not analysed total DNAPK and CHK1 protein levels in every single experiment because total protein expression is not significantly affected by AZD1775 treatment, in stark contrast to the increased phosphorylation of DNAPK/CHK1 by AZD1775. This is now more clearly stated in the text. Expression of total CHK1 protein was analyzed in several of the previous figures. Total DNA-PK protein levels is analyzed in new Figure 4—figure supplement 1D and Figure 5—figure supplement 2B.

Figure 1A – The high content imaging screen measured cell numbers.How is it related to the curve on the right (which is a crystal violet assay)?

It is correct that the high-content imaging (HCI) screen measure changes in total cell number at single-cell resolution, producing dose response curves and drug response characteristics, while the response curve to the right is drug response quantified by crystal violet assay. We apologize for the confusion in this case and have exchanged the crystal violet data with a drug response measurement from the HCI screen. The crystal violet data (which served as validation of HCI data) is now moved to Figure 1—figure supplement 1B. To clarify, the HCI data evaluates IC50 – concentration of compound achieving 50% reduction in cell number, EC50 – concentration of compound achieving 50% of the maximum effect, AA – Activity Area. This information is now added to the schematics in Figure 1A and legend, and the response data shown in new Supplementary file 1 (addressing this point by reviewer #1 and comment by reviewer #3 below).

Figure 1C – this supposedly classified cell lines based on synergy between AZD1775 and AZD6738 – it is not clear how synergy was defined.Why were the concentrations of the drugs at 500nM and 1000nM respectively? This figure arbitrarily separated cell lines based on two fixed drug concentrations. Why is MDA-MB-231 not included?I also disliked with the term "recurrent". It implies recurrence in a clinical setting.How was synergy determined?The combination was asserted to be synergistic but what model was used and what were the synergy scores for the combinations?Also never define what the statistical significance of **** is or what test was used.What are the different IC50 values for these cell lines?A table is needed here.

“Recurrent” is now changed to “recovering”.

Referring to the previous point (IC50), additional drug response data is provided in Supplementary file 1 and statistical significance more clearly defined in Figure 1C legend.

Synergy can be defined as a combination effect that is greater than the additive effect expected from the knowledge of the individual drugs. As a single agent, 1 μm of AZD6738 has minimal inhibitory effect in all the cell lines (mean ± SD = 1918 nM ± 662 nM, p<0,05) and acute synergy of the combination was therefore analyzed at a fixed ratio of 1 μm AZD6738 and 500 nM AZD1775. Synergy was calculated by ¨Combination Index” (CI) using CImbinator, a publicly-available online drug-combination calculation tool (Flobak et al., 2017), based on the least-squares method proposed by Dr. Ting-Chao Chou (Chou, 2010). Synergy scores are provided in the Figure 1C legend. As mentioned in the text, acute synergy between AZD1775 + AZD6738 was recently published while this study was ongoing. Therefore, we chose to focus our resources on experiments that would cover new ground and increase the impact of the study. Nevertheless, to accommodate reviewer #1’s concern, we now provide more in-depth synergy analysis (analyzing synergy at different drug concentrations) for AZD1775 + AZD6738 (new Figure 1—figure supplement 1C and legend, described in the main text). In addition, we also provide new data demonstrating synergistic interaction between AZD1775 + DNAPK inhibitor (NU7441) combination (new Figure 4c and Figure 4—figure supplement 1B and legend, described in the main text). Synergy scores were calculated based on the Bliss model using SynergyFinder, a specialized online tool (Ianevski et al., 2017).Figure 1E is described as clonogenic assays but it cannot be since it is only 7 days. No colony formation is possible in 7 days.No description in Materials and methods. Quantification?What are the ODsfor all of the different wells, 2 of the recurrent cell lines – BT20 and HCC1143 look like the mono treatment was also inhibiting growth a decent amount compared to the DMSO control, so what percentages of cells were growing post treatment?We apologize for the inadequate description of the assay and are thankful that this was pointed out. All colony assays in the manuscript are 2D crystal violet assays for adherent cells. A more detailed description of the assay is now included in the Materials and methods section and described in the corresponding legends. Further, quantification of cell numbers following drug washout (recovery of proliferation) is now more clearly described in the main text, and new Figure 1—figure supplement 1D and corresponding legend.Figure 1F – the flow cytometry data is very difficult to read (too small).Why does the BT20 cell line has no to little γH2AX foci when clearly the combination treatment in Figure 1C resulted in over 50% cell death.

Sorry for the confusion in this case. With respect to Figure 1F, AZD1775 mono- and the combination-treatment in fact increase γH2AX in BT20 cells (13% and 19%, respectively). The important point is that AZD1775 induce little γH2AX in replicating DNA in these cells (2,4%, as measured by EdU pulse-labeling) as compared to hypersensitive HCC38 cells (27,6% EdU/γH2AX+). Figure 1C measure the overall inhibitory effect (combined cytostatic and cytotoxic effects). This is now described clearer in the legend to Figure 1C. To further emphasize the results in Figure 1F, we include graphs showing the % of γH2AX+ in both the EdU-negative as well as the EdU-positive population from one representative FACS experiment.

Figure 3B – Statistics?Did n=4 for control and each mono treatment and then n=6 for combo so can you get proper stats from n=4?

The statistics are based on tumor growth index i.e. the relative increase in tumor volume at a particular time point compared with the tumor volume at the start of treatment for each tumor. This is now explained in the legend to Figure 3B. Analysis from 4 vs 6 mice gave significant differences between the two groups for most but not all time points using this method. We should point out that the Swedish authorities are very strict with the 3R rules and we therefore try to keep the number of mice to a minimum in each experiment.

Images of higher magnification are provided, including scale bars with reference size.

Western blots need to be quantitated (Figure 3G, Figure 4C, D, E and F, Figure 5D, Figure 6B, C, F and G). Statistics method needs to be clearly described.Raw data for Figure 2—figure supplement 1 needs to be provided.

We now provide quantification of representative immunoblots, where relevant. We also provide quantification of immunofluorescence data in Figure 5C. Statistical methods are clearly described in Materials and methods section. Raw data for Figure 2—figure supplement 1 is provided as Figure 2—source data 1.

Assuming that 500nM AZD1775 was used throughout the manuscript, it produces variable results. The inconsistencies need to be addressed.

We disagree, 500 nM AZD1775 produce consistent results (for example, AZD1775 clearly increase phosphorylation of CHK1, DNAPK and γH2AX, and reduce phosphorylation of CDK1/2 in multiple blots) although the amount of increase/decrease vary depending on the treatment time for certain experiments. It is correct that we generally use 500nM AZD1775 for biochemical experiments (close to IC50 in MDA-MB-231 cells) but different concentrations are used in drug response experiments.

The authors suggest that the increased γH2AX is caused by increased Cdk2 activity as evident by the fact that the CDK inhibitor Milciclib reduces γH2AX levels.Granted, Milciclib is slightly more selective towards Cdk2, but Milciclib also inhibits Cdk1 as well as other CDKs.Given that increased Cdk1 activity is shown to induce chromosome pulverization in WEE1 inhibited cells (Duda et al., Developmental cell, 2016; Lewis et al., Cancer Research, 2019) it is unlikely that the reduced DNA damage observed with Milciclib treatment is solely dependent on loss of Cdk2 activity.

Since Milciclib inhibits CDK2 at a 9-fold lower concentration than CDK1, it is likely that the main effect on γH2AX goes through CDK2, which is also compatible with our other observations throughout the manuscript. Nevertheless, since we cannot exclude that the effect of Milciclib in part goes via effects on other CDKs, we have modified the text accordingly.

Increased AKT/mTOR signalling is reported to induce AZD1775 acquired resistance through increase nuclear activation of Chk1 (Sen et al., 2017).The authors show that the loss of PTEN increases AKT signalling consistent with the role of PTEN in regulating PIP3 levels, but inconsistent with the Sen et al. study.The authors should cite this paper and discuss the potential discrepancy.

We believe the reviewer has misunderstood. First of all, Sen et al. report that CHK1 is activated in AZD1775-resistant cells as an escape mechanism to overcome accumulation of DNA damage and cell death. Our findings are not inconsistent with this result, as a matter of fact, we show that CHK1 is phosphorylated in response to AZD1775 treatment and also demonstrate that DNAPK is critical for CHK1 activation when ATR is inhibited (Figure 5). AZD1775 induce the replication checkpoint arrest (phosphorylation of CHK1 and downregulation cyclin E expression), followed by checkpoint recovery and proliferation post-treatment (Figure 3). Second, Sen et al. studied acquired resistance in SCLC models and showed that high AXL/MET/mTOR expression is associated with resistance to AZD1775, while our work focused on identification of factors regulating AZD1775 sensitivity and recovery post treatment (breast cancer model). The important point here is that our study reveals that loss of PTEN is associated with increased sensitivity to AZD1775, presumably due to an impaired checkpoint activation (reduced pS345-CHK1 and elevated cyclin E levels, Figure 6) resulting in replication-induced DNA damage and cell death in PTEN-deficient cells (Figure 2). Although not formally proven in our study, loss of PTEN and hyperactivation of AKT may impair CHK1 activation in response to AZD1775 treatment (reduced pS345-CHK1, Figure 6). Notably, Puc et al., 2005 showed that loss of PTEN and activation of AKT leads to reduced nuclear CHK1 localization promoting genomic instability in tumor cells. Puc et al. traced the checkpoint defect in PTEN-negative cells to AKT and phosphorylation of CHK1 at serine 280 leading to cytoplasmic retention of CHK1. This result is inconsistent with Sen et al., who suggest that activation of AXL/mTOR induces CHK1 accumulation in the nucleus. In summary, we found reduced CHK1 phosphorylation/activity and increased sensitivity to AZD1775 in PTEN-deficient cells, similar to Puc et al. At this stage, it is unclear whether increased AKT activity in PTEN-deficient cells is the primary cause for the compromised CHK1 activation (reduced pS345-CHK1) and deregulated replication checkpoint (elevated cyclin E levels) in response to AZD1775. We now discuss and cite both of these papers.

Given that the authors claim that the loss of PTEN and DNAPK enhance AZD1775 sensitivity, the authors should confirm that PTEN and DNAPK re-expression in KO cells restores AZD1775 resistance.

We have now restored PTEN expression in PTEN-negative, AZD1775 hypersensitive HCC1937 cells. As shown in new Figure 2G, expression of PTEN significantly reduced sensitivity to AZD1775 monotherapy, strongly supporting a critical function for PTEN in suppressing excessive replication stress induced by AZD1775. In addition, we show that reintroduction of PTEN in PTEN-negative cells promote cellular recovery post AZD1775 treatment (new Figure 2H).

Restoring DNAPK in DNAPK-KO background is a formidable task that would take considerable time before it produced any tangible results and is therefore beyond the scope of the present study.

Provide raw data for the graph presented in Figure 2E.

Done.

Rectified.

The authors asserted there is significance many times without showing the statistics for the significance. The authors asserted that the combination is synergistic in the cells they tested it in however don't show the synergy score or say how they determined if it was synergistic. The authors frequently asserted growth was decreased or didn't grow again however rarely show quantification of their crystal violet assay for how many cells actually grew.

Sorry this was not clearer. Statistical significance is now provided when relevant, all biochemical experiments have been repeated at least three times (if not otherwise stated) and representative blots/FACS data are presented. Regarding synergy and recovery of proliferation, see above response to point – Figure 1C and 1E. As described, we provide new data (quantification of cell recovery by crystal violet assay) in Figure 1—figure supplement 1D and corresponding legend.

Reviewer #1 (Significance):While themanuscript is potentially interesting and significant, it isnot interpretable at the current state.

We thank the reviewer for the detailed review and the opportunity to bring in new important data to improve our manuscript.

Reviewer #2 (Evidence, reproducibility and clarity):In this study, Suryo Rahmanto and colleagues investigate the molecular mechanisms underlying the initial response and subsequent recovery of basal-like breast cancer cells following AZD1775 therapy.They show that inhibition of WEE1 by AZD1775 leads to DNA-PK phosphorylation and cyclin E destruction thus shielding basal-like breast cancer cells from replication-induced DNA damage. The authors carried out a systematic analysis of AZD1775 sensitivity and resistance and identified a group of basal-like breast cancer cells hypersensitive to AZD1775 that are characterized by loss of PTEN.In my opinion, the experiments are presented in a straightforward and logical fashion. The data presented support the authors' conclusions.Reviewer #2 (Significance):This is an interesting study that demonstrates that PTEN is a potential predictive marker of AZD1775 response indicating that targeting WEE1 may represent a promising therapy for PTEN-deficient breast cancers.I find the manuscript of considerable interest for a broad as well as cancer biology readership.

We appreciate that the reviewer find our study interesting and relevant for the broad readership and acknowledge the feedback that results are well-presented, with supporting conclusions.

Reviewer #3 (Evidence, reproducibility and clarity):Summary:The authors analysed breast cancer cell line responses to inhibition of WEE1 kinase by AZD1775. They test a panel of breast cancer (BC) cell lines with global proteome/transcriptome profiling. This led to identification of two response groups of basal-like BC cell lines distinguished by PTEN status. High levels were resistant and PTEN low were sensitive to WEE1i. The molecular mechanism underlying the PTEN role was not fully elucidated. Overall, this suggests that PTEN might be a biomarker for stratification of breast cancer patients for more efficient WEE1 therapy. The authors then show that DNA-PK inhibition also sensitises to WEE1i, which was linked with DNA-PKi suppressing CHK1 activation.Major points:AZD1775 toxic effects are exerted in S and G2 cell cycle phases. For this reason, it is crucial to properly evaluate that there are no major cell cycle population differences between the different cells lines and if there are, factor them in into the analysis.This appears not accounted for in their analysis (at least I did not find it in the Materials and methods section or explicitly stated in the text).

This is a very relevant point and we thank the reviewer for the opportunity to clarify this in more detail. We provide additional information demonstrating that there are no major differences in cell cycle populations/proliferation between the cell lines that significantly influence the response or recovery following AZD1775 treatment, see new Supplementary file 1 and new Figure 6—figure supplement 1G. This is now clearly described in the legends and in the main text. Furthermore, we have analyzed expression of proteins related to proliferation and find no major difference between the basal-like cell lines, regardless of AZD1775 sensitivity/response. As expected, a difference in proliferation-related proteins is observed when comparing luminal and basal-like cells (new Figure 6—figure supplement 1E and F).

The authors identified factors that are correlated with resistance (such as CDC7), but do not comment on them at all.Also, TP53I11 is completely new factor that has not been investigated at all, which could be an interesting follow up project.

We agree with the reviewer that other factors associated with increased or reduced AZD1775 sensitivity would be interesting to follow up. To investigate this further, we analysed the expression of CDC7, TP53I11 and CTPS1 in the different cell lines and provide protein data in new Figure 2—figure supplement 2A. Although linked to AZD1775 sensitivity, the main discriminator of these markers is the molecular subtype, basal-like vs luminal cells (which is not the case for PTEN which also differentiate between basal cells, Figure 2 and 6E). Regardless, loss of PTEN and activation of dormant origins during replication stress (He et al., 2015) supports an increased dependency on replication initiating factors such as CDC7 and cyclin E which is in line with our findings.

Authors spend a significant amount of the paper focusing on the efficacy of the double treatment WEE1i + ATRi, which could be reduced considering they have more novel findings.

As the AZD1775 + AZD6738 combination was published when this study was ongoing, we have focused on the more novel findings (DNAPK/PTEN), however, the AZD1775 + AZD6738 combination drug washout data are still relevant as the previous studies were not primarily addressing recovery post treatment. If the reviewer insists, we can certainly reduce the amount of data related to the acute combined inhibition of WEE1 + ATR.

There a factual errors and citations that are absent do not match the point mad – i.e. page 16 line 359 and citation also does not fit their statement.Sometimes citations are missing from some of the claims, such as Page 16 line 367.Where are references 42-49?

We apologize, our mistake. This is now corrected, and references provided.

Figure 1F, in the FACS profile one can already see that the S phase population in HCC30 is larger than in the other cell line.Bukhari et al., quite clearly shows this. The authors have it in the reference list, but it is not cited.

We apologize for the confusion. The BT20 and HCC38 cells in fact have similar S phase population (BT20 – 37% EdU+, HCC38 – 38% EdU+). The point is that AZD1775 induces little γH2AX in replicating DNA in BT20 cells (2,4%, as measured by EdU pulse-labeling) as compared to hypersensitive HCC38 cells (27,6% γH2AX+/EdU+ cells). As this was obviously not clear in the previous figure, we now include γH2AX data for both the EdU-negative as well as the EdU-positive population.

There are multiple cases where statistics is missing on the figure panels.

Statistics is now provided when relevant. As stated in the legend, all experiments were repeated at least three times if not otherwise stated.

Reviewer #3 (Significance):The authors uncover novel potential biomarker (PTEN) and a role for DNA-PK in tolerance to WEE1i.The suggested modes of actions are in line with the consensus in the field that factors that impair checkpoint or replication fork stability during replication stress have synergistic effects with AZD1775. Thus,the manuscript has interesting findings (mainly on PTEN) and has aspects that might eventually be translated into a clinical setting, this is also highlighted and discussed in the manuscript.It would be very interesting if the authors follow up the PTEN aspects more vigorously, which should include inhibiting key factors on the PTEN pathway.Next, authors can test hypothesis based on their findings and the substantial literature as also discussed in the current version of the manuscript.

We thank the reviewer for the feedback. We now provide new data demonstrating that restoring PTEN expression in AZD1775 hypersensitive, PTEN-negative HCC1937 cells, significantly reduce AZD1775 sensitivity (new Figure 2G) and promote recovery post treatment (new Figure 2H).

The authors are not citing important papers in the field such as the Aarts papers from 2012 and 2015 (especially Aarts et al., 2015, which did a WEE1i synthetic lethality screen).This appears due to a major reference list error where a number of references are missing.

Our mistake, the original manuscript in fact included these references but for unknown reason was lost during the submission process. This has now been corrected and we reference both these papers in the new version of the manuscript.

We thank the reviewer for relevant comments and for the opportunity to modify our manuscript accordingly.